# Simulation of cold powder snow avalanches considering daily snowpack and weather situations

Julia Glaus[1,2], Katreen Wikstrom Jones[3], Perry Bartelt[1,2], Marc Christen[1,2], Lukas Stoffel[1], Johan Gaume[1,2,4], and Yves Bühler[1,2]

[1]WSL Institute for Snow and Avalanche Research SLF, Davos Dorf, 7260 Switzerland
[2]Climate Change, Extremes, and Natural Hazards in Alpine Regions Research Centre CERC, Davos Dorf, 7260 Switzerland
[3]Alaska Division of Geological & Geophysical Surveys, Anchorage
[4]Institute for Geotechnical Engineering, ETH Zurich, Zurich, 8057 Switzerland

**Correspondence:** Julia Glaus (julia.glaus@slf.ch)

**Abstract.** Snow avalanches are rapid gravitational mass movements that pose a significant hazard to both humans and infrastructure, including traffic lines. Risk management in mountainous regions usually relies on experience of avalanche experts, observations in the field, weather and snowpack measurements and numerical simulations.

Ensuring road safety requires considering daily weather conditions, snowpack characteristics, and terrain features. To include a numerical model in the decision-making process for road safety, it is essential to incorporate all these factors and utilize in-situ measurements as input parameters for the simulations.

This study investigates the predictive capabilities of the numerical simulation model RAMMS::EXTENDED, an extended version of the well established RAMMS software developed at the WSL Institute for Snow and Avalanche research SLF over the past fifteen years, to estimate avalanche run-out distances along an important infrastructure corridor in the Dischma Valley near Davos, Switzerland. Specifically tailored to cold powder avalanches dynamics, taking into account the temperature of the snowpack and entrainment, our inquiry utilises meteorological station measurements as an input to evaluate the model's performance.

In this paper, we begin by providing an overview of the model, examining its physical and practical aspects. We then conduct a sensitivity analysis on input and system parameters, focusing on avalanche dynamics representation. Leveraging drone-based observational data, we perform a comparative analysis to validate the simulation results.

In addition to recalculating avalanches due to the sensitivity analysis, we show that we achieve meaningful predictions of the avalanche run-out distance for cold powder avalanches by incorporating snow height and snow temperature measured by weather stations at two different altitudes near the avalanche release zone. In the future, a refined version of this approach could allow for near real-time hazard assessments which has the potential to significantly improve the decision-making protocol for road closures and re-openings. Furthermore, we plan to calibrate the model for wet-snow avalanches to cover a larger range of weather and snowpack scenarios.

# 1 Introduction

Nowadays, in some regions, a robust network of measurement stations provides point information on snow depth, temperature, and wind speed. In this publication, we aim to explore how these data sources can be leveraged to simulate avalanche runouts to represent daily conditions. Using the example of road safety, we will demonstrate how this information can be applied to protect infrastructure. The same approach is adaptable for other safety frameworks, such as ski slopes or buildings in avalanche prone areas.

Due to economic and environmental constraints, many mountain roads cannot be effectively protected using long-term technical measures to prevent avalanche release (avalanche defence structures) or inundation (road alignment, snow sheds and tunnels). Therefore local hazard experts must make decisions to close roads and stop all traffic during avalanching periods. These decisions rely on information from the warning services, the interpretation of measurement data and experience (Stoffel and Schweizer, 2008). Increasingly computer-based expert systems such as the nearest-neighbour model for regional avalanche forecasting called NXD (Brabec and Meister, 2001) or AI systems are applied to help the hazard experts.

In this work, we focus on adding information to the decision-making process by combining data from weather stations, numerical modelling and drone measurements. The primary goal is to answer the question whether an avalanche could reach a road under specific snowpack and weather conditions. Having this information at hand could enhance road safety mitigation and reduce the road closure times to a minimum. For this approach to be successful and to include all possible avalanche paths along a road, accurate reports of snowpack and weather conditions are needed. This data must be collected as close as possible to the specific avalanche path.

The utilisation of numerical avalanche dynamics modelling to enhance road safety signifies an important paradigm shift in avalanche engineering. While numerical models have traditionally and extensively been adopted for generating hazard maps and designing avalanche defence structures along specific avalanche paths, they often don't include crucial snow properties such as snow cover layering, density, temperature moisture content to represent daily conditions. Avalanche fracture heights are typically determined through statistical analysis of long-term snow accumulation data from measurement stations (Salm et al., 1990). Following an approach pioneered by Voellmy (Voellmy, 1955) extreme avalanche events are typically addressed using calibrated parameters derived from historical avalanche occurrences (Gruber and Bartelt, 2007). While this approach is suitable for hazard mapping, it fails to leverage recent advancements in automatic weather stations or drone measurements (Bühler et al., 2017). Consequently, the output from numerical models available to local hazard engineers for deciding whether to close a road is limited, as it becomes challenging to correlate specific measured data with potential avalanche runouts.

A first system using numerical simulations for road safety is implemented in Chile as described in (Valero et al., 2018). It uses avalanche dynamic modelling based on RAMMS to predict whether an avalanche reaches a road. For the input data it relies additionally on the simulation tool SNOWPACK (Cerda et al., 2016; Lehning et al., 1999).

As we experiment with new applications and develop more complicated modelling chains, we also place new demands on these existing numerical models. To accurately represent snow and weather conditions, a model must have the capability to encompass avalanches with different flow regimes (including wet-, mixed- and dry snow avalanches), consider snowcover

entrainment and mass growth, the braking effects of different forest compositions and, most importantly, the influence of snow temperature. Existing avalanche dynamics models which focus on the flowing regime, ELBA (Keiler et al., 2006), OpenFOAM (Rauter et al., 2018), SAMOS-AT (Sampl and Granig, 2009), Avaframe (ava, 2023) and RAMMS::Avalanche (Christen et al., 2010) meet only parts of these requirements.

In this paper, we utilise three well-documented avalanches that overflowed a mountain road near Davos (Switzerland) to investigate how avalanche dynamics models can effectively be used with weather station data. The avalanches were artificially released and developed into a mixed flowing powder type. Post-event drone scans provided detailed information of runout and snowcover distribution. We apply an extended RAMMS model that includes snow temperature (Valero et al., 2016), entrainment (Bartelt et al., 2018) and formation and propagation of the powder cloud (Zhuang et al., 2023a). The model was calibrated using avalanches observed at Vallée de la Sionne (VdlS), considering only those that did not reach the counter slope. In the first part of this paper, we provide an overview of the current version of the model, summarize recent literature on RAMMS::EXTENDED, and present the current calibrations. In the second part, we back-calculate the observed avalanches using temperature data from nearby snow monitoring stations and the parameter set from VdlS. We show how the model reacts to changing boundary conditions and the sensitivity of model performance on variation of parameters. Our results highlight the challenges of using avalanche dynamics models for road safety applications.

## 2 Observations and Methods

### 2.1 Avalanche events, Davos, 15 January 2019

We examine three separate avalanche events that took place in the vicinity of Davos, Switzerland. These incidents occurred in mid-January 2019 in the Dischma valley during a cold weather period, leading to the formation of mixed flowing-powder avalanches, see Figure 1. The three avalanche tracks are located on the northeastern slope of Brämabühl (Davos) and have the names *Wildi*, *Rüchi* and *Chaiseren*. The release zones are all located at roughly 2300 m.a.s.l on a northeastern aspect. The tracks drop between 650 m and 750 m in elevation, running over a well-used cross country skiing track and local a road. The tracks are somewhat channelised below the release zone, but they open to wide, laterally unconstrained runout zones at the valley bottom. In the past, avalanches from these tracks have blocked the road connecting the inhabitants of the valley and Davos, wooden buildings have been destroyed and trees in the surrounding forests have been blown-over by avalanche air-blasts. For all three tracks hazard maps exist.

The 2019 avalanche events are unique since substantial snowfall preceded the cold spell, with snow depths as high as 2.5 m measured at nearby snowcover monitoring stations. A strong winter storm passed through the valley on the 14 January 2019 with strong winds which redistributed snow on the slope (Glaus et al., 2024). The avalanches eroded a deep, cold snowpack, which contributed to the formation of powder avalanches. The avalanches were artificially released and additional pictures from the helicopter during the avalanche control provided estimates of the powder cloud speed (around 30 m/s in the runout zone) and height (approximately 40 m). For the Rüchi and Chaiseren path, the approximate cloud speed can be estimated by analysing the position of the powder cloud front over time as illustrated in Figure 2. The avalanches considerably increased in

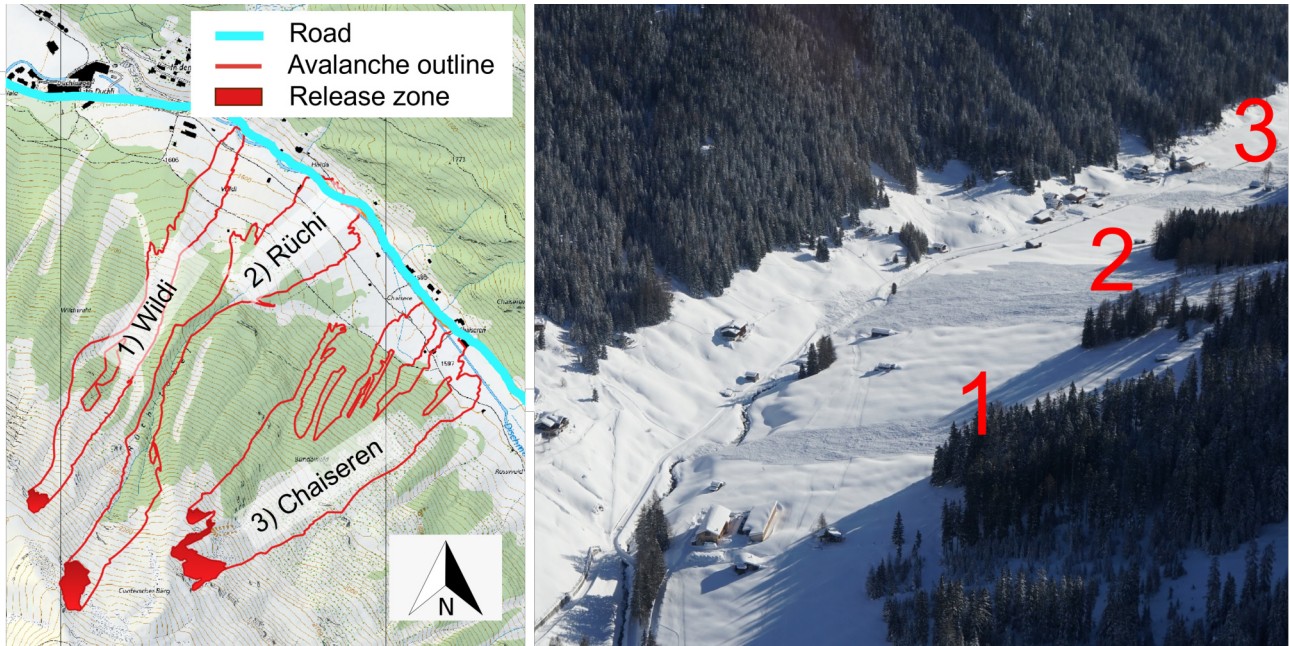

**Figure 1.** On the left side, depiction of the measured release zones of the three avalanche paths 1. Wildi, 2. Rüchi and 3. Chaiseren. The measured outline of the three avalanches at Brämabühl 15 January 2019 from the drone data are marked in read (map source: Federal Office of Topography). On the right side, the deposition of the three powder avalanches in the Dischma valley (Davos, Switzerland) originated from north east facing slopes. Powder avalanches often reach the valley road after traversing a flat runout zone (Pictures made by Vali Meier, SOS Davos Klosters). The grid lines show a distance of 1km.

mass after release due to snow entrainment. In the following days, a field campaign was carried out by the institute for snow and avalanche (SLF) to gather data concerning the location of the release zones, entrainment heights and avalanche runout lengths. The data was coupled with temperature data from nearby snow monitoring stations (Weissfluhjoch and IMIS SLF2) which are situated 6 km and 3 km away. The Weissfluhjoch station (2536 m.a.s.l) is located roughly at the same elevation as the avalanche release zones and the IMIS SLF2 at the same elevation as the runout zones (1570 m.a.s.l).

On-site data collection was conducted with structure-from-motion photogrammetry (Bühler et al., 2011) using drones, allowing the measurement of snow heights by comparing post-avalanche elevation surfaces with bare ground surfaces during the summer obtained from the Federal Swiss topographical survey (SwissTopo) (Swisstopo, 2024). in Figure 3, the measured post-avalanche snow heights for all three avalanche tracks are shown. This drone data allowed the delineation of the release zones, thanks to a clearly visible stauchwall. Avalanche fracture height ($d_0$) perpendicularly to the terrain could be estimated by comparing measured snow heights at similar altitudes to the average snow height in the release zone after avalanche release. Additionally, from the drone measurements the snow distribution gradient $\nabla D$ could be determined. These values are reported as the average decrease in snowcover height per 100 m drop in elevation (Figure 6).

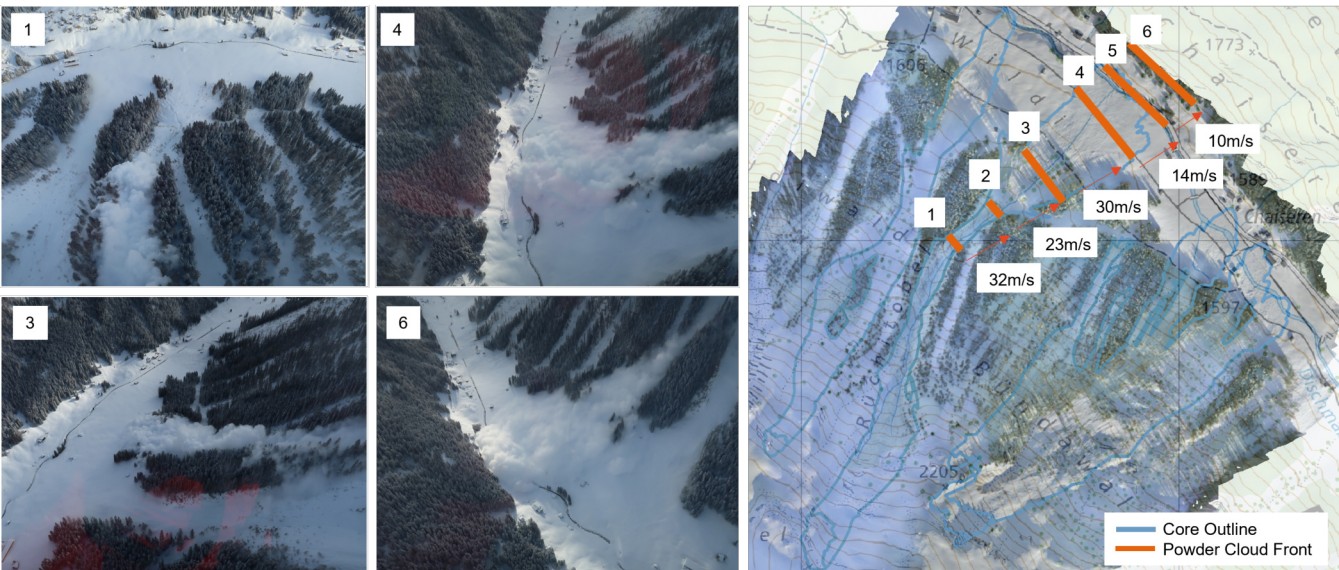

**Figure 2.** Estimation of the powder cloud velocity from the helicopter-captured images for the Rüchi path. The velocity was calculated by comparing the distance travelled by the cloud to the time interval between the images. (Pictures made by Vali Meier, SOS Davos Klosters)

To better understand temperature gradients $\nabla T$ along the path (Figure 6), we analysed snow pits concurrently with temperature readings at Weissfluhjoch and SLF stations (Attachment A1). The average temperature of the released snowpack was used to infer temperature gradients and snow density in the release zone, further interpolated across stations at varying altitudes to understand snow cover gradients.

Our approach was to simulate the avalanche events from 2019 based on meteorological data and snowpits and validate the results to the avalanche outlines and dimensions that we measured based on the post-avalanche drone data. We started by simulating one avalanche and then apply the parameter set of this avalanche to the other two tracks as they were triggered almost at the same time and hence should have the same input data. A summary of the snowcover and temperature input data of the Brämabühl events is presented in Table 1. The same avalanching period in January 2019 produced a well-documented event on the nearby Salezer avalanche track in Davos (at the same day) as well as a powder avalanche at the experimental VdlS test site (Ammann, 1999) which we could additionally use for validation. In this publication we keep the focus on the Brämabühl event to describe our methodology.

## 2.2 Evaluation Method

Our evaluation approach was focused on determining if an avalanche can reach the road. We also estimated the extent of both the dense avalanche core and powder cloud impact pressures. To do so, we have developed a post-processing tool to assess model outputs based on the maximum values per pixel of velocity, flow height, and pressure per calculation cell as described in (Glaus et al., 2023). We extract the outlines of both the core and cloud, determining the longest distance by identifying

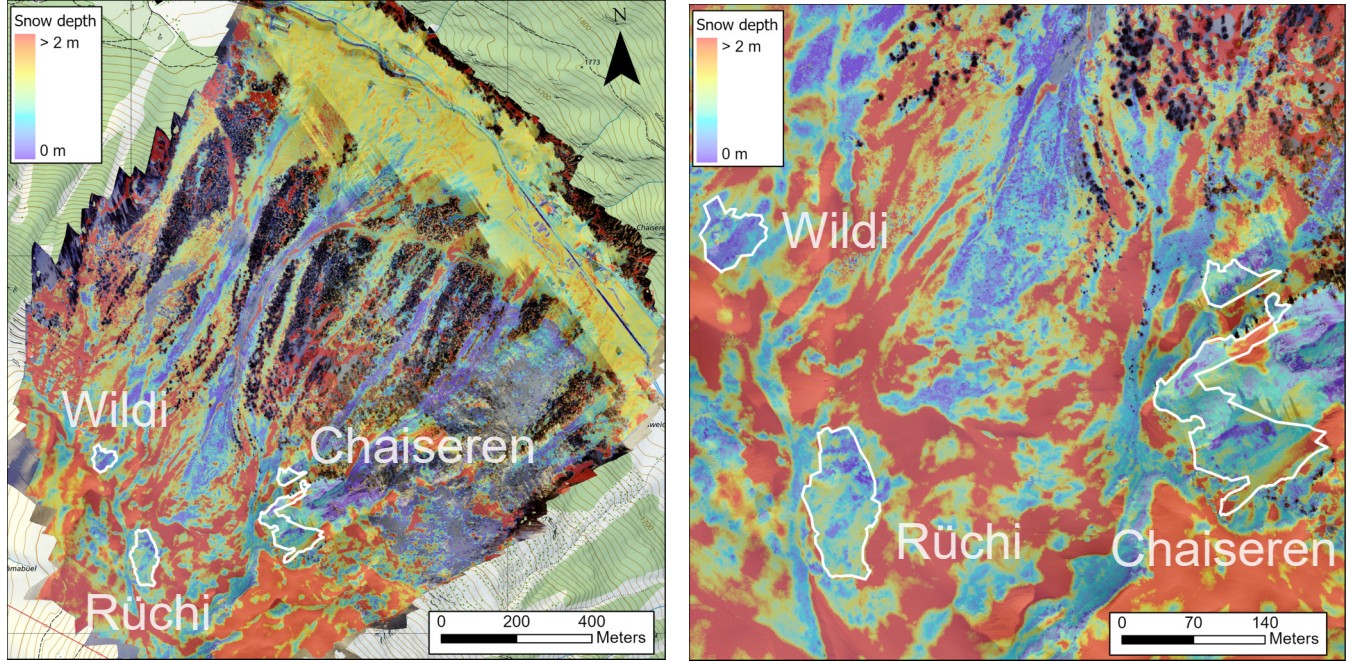

**Figure 3.** Snow depth distribution mapped photogrammetrically with the eBee RTK drone on 15 January 2019 for the entire area (left) and zoomed to the avalanche release zone (right) after the avalanches got triggered. Significant wind redistribution effects are visible in the image (map source: Federal Office of Topography).

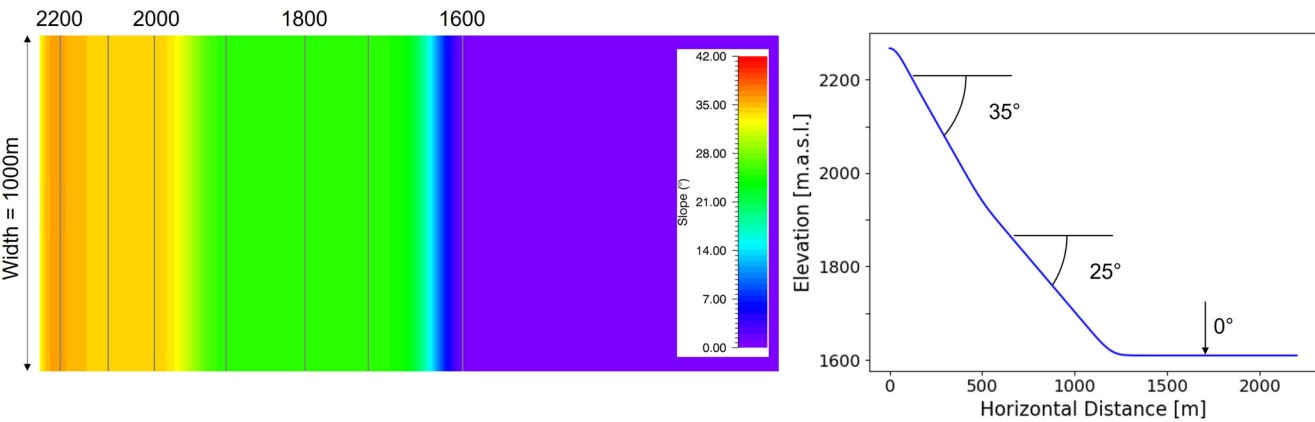

**Figure 4.** Visualization of the idealized plane inspired by the steepness of the Rüchi avalanche path. Left: Bird's-eye view of the plane with contour lines and altitude marked. The colour gradient indicates slope angles, ranging from low (flat areas) to high (steep areas). Right: Cross-section of the plane, illustrating the elevation profile along a selected transect.

the two most distant points using the Convex Hull algorithm combined with the Rotating Calipers method (in Python via scipy.spatial.ConvexHull). The resulting avalanche runout distances for the Brämabühl event are marked in Fi. 13 and the values are given in Table 1. For core outline we identify pixels with a flow height greater than 0.1 m and a velocity less than 1 m/s. For the cloud, the outline is based on the stagnation pressure with a lower threshold set at 0.5 kPa computed by $0.5\rho_\Phi u_\Phi^2$ with the airblast profile described in Zhuang et al. (2023a). While this method works well for simple avalanches, it requires careful consideration in cases where avalanches exhibit finger formation or the avalanche strongly deviates in the lateral direction.

We conducted a sensitivity analysis to determine how the avalanche responds to variations in the initial conditions and model parameters. First we back-calculate the avalanches with the given parameter set from the VdlS calibration. We vary one input parameter at once and comparing it to the observed avalanche outlines which are presented in Figure 1. To set the input parameter set, we used the weather station data and systematically varied one input parameter at a time, with an emphasis on parameters that practitioners can measure. The goal of that study was to quantify the impact of uncertainties in input parameters on simulation outcomes. Subsequently, we expanded to varying pairs of parameters, such as snowcover temperature and temperature gradient. Details of these initial findings are discussed in (Glaus et al., 2023).

We conducted a sensitivity analysis to determine how the avalanche responds to variations in the initial conditions and model parameters. Given the complexity of avalanches, a large number of model parameters required calibration, but not all can be presented in detail in this publication due to space limitations.

First we back-calculate the avalanches with the given parameter set from the VdlS calibration for all the hardcoded parameters. To set the input parameter set, we used the weather station data and systematically varied one input parameter at a time, with an emphasis on parameters that practitioners can measure. The results we compare to the observed avalanche outlines which are presented in Figure 1. The goal of that study was to quantify the impact of uncertainties in input parameters on simulation outcomes. Subsequently, we expanded to varying pairs of parameters, such as snowcover temperature and temperature gradient. Details of these initial findings are discussed in (Glaus et al., 2023).

## 2.3 A Method to Model Snowcover Distribution $d_\Sigma(Z)$ and Temperature $T_\Sigma(Z)$

The underlying idea behind the SLF procedure on avalanche dynamics calculations is to exploit long-term, measured frequency-magnitude snowfall data to determine avalanche fracture heights $d_0$. Avalanche fracture heights are explicitly related to measured, extreme three-day snow depth increase (Salm et al., 1990). This procedure underscores two salient assumptions of the Swiss guidelines. Firstly, regional variations in snowfall climatology are included via the measurement data (snow height frequency) and secondly, extreme avalanche activity is directly related to intense new snowfall. In the following we develop a methodology to determine avalanche entrainment heights for road safety calculations within the framework of these existing Swiss guideline procedures, that upholds these two basic assumptions.

Because the snow monitoring stations are located at different elevations and different slope angles, snow accumulation data must be adjusted to account for the specific elevation and slope of the avalanche release zone. Within the guidelines this is performed by applying a snow height gradient $\nabla D$. If $Z_m$ is the elevation of the measurement station, and $Z_0$ is the elevation

**Table 1.** Overview of the input data for snow height and temperature data for the Brämabühl events.

| Snow cover disposition | Wildi | Rüchi | Chaiseren | Source |
|---|---|---|---|---|
| Release height $d_0$ | 0.95 m | 1.45 m | 0.95 m | Drone |
| Release density $\rho_0$ | 193 kg/m$^3$ | 193 kg/m$^3$ | 193 kg/m$^3$ | SNOWPACK |
| Maximum erosion height $d_0^*$ | 1.15 m | 1.85 m | 1.15 m | Drone |
| Erosion gradient $\nabla D$ | 0.1 m/100m | 0.1 m/100m | 0.1 m/100m | SNOWPACK |
| Erosion density $\rho_\Sigma$ | 193 kg/m$^3$ | 193 kg/m$^3$ | 193 kg/m$^3$ | SNOWPACK |
| Release temperature T | -8.1 °C | -8.4 °C | -7.8 °C | SNOWPACK |
| Temperature gradient $\nabla T$ | 0.5 °C/100 m | 0.5 °C/100 m | 0.5 °C/100 m | SNOWPACK |
| Runout distance | 1340 m | 1545 m | 1160 m | Drone |
| Grid resolution | 5 m | 5m | 5 m | - |

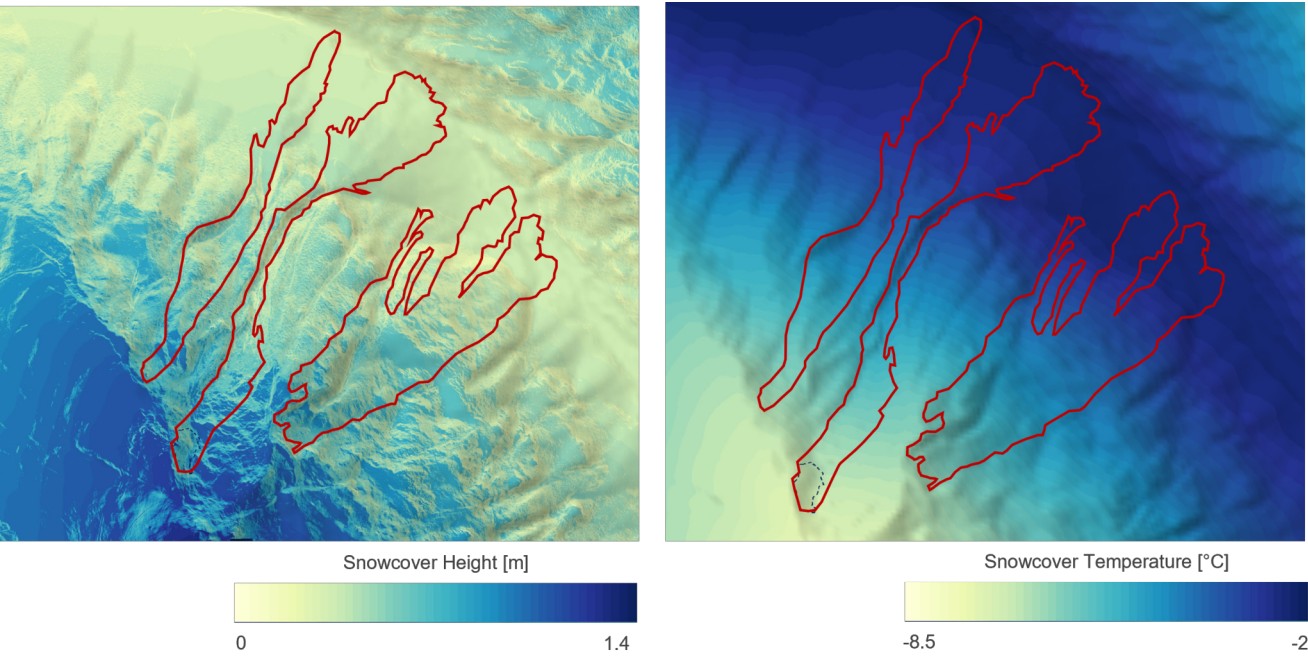

**Figure 5.** Simulated erodable snow cover (left) and temperature distribution (right) applied for the modelling In red the observed outlines from the Brämabühl events are marked.

of the avalanche release zone, the first iteration of the fracture height $d_0^{(1)}$ is found by adjusting the value obtained $d_m$ obtained from the statistical frequency-magnitude analysis of the measurement station (Salm et al., 1990),

155
$$d_0^{(1)} = d_m + \nabla D \left( Z_0 - Z_m \right). \tag{1}$$

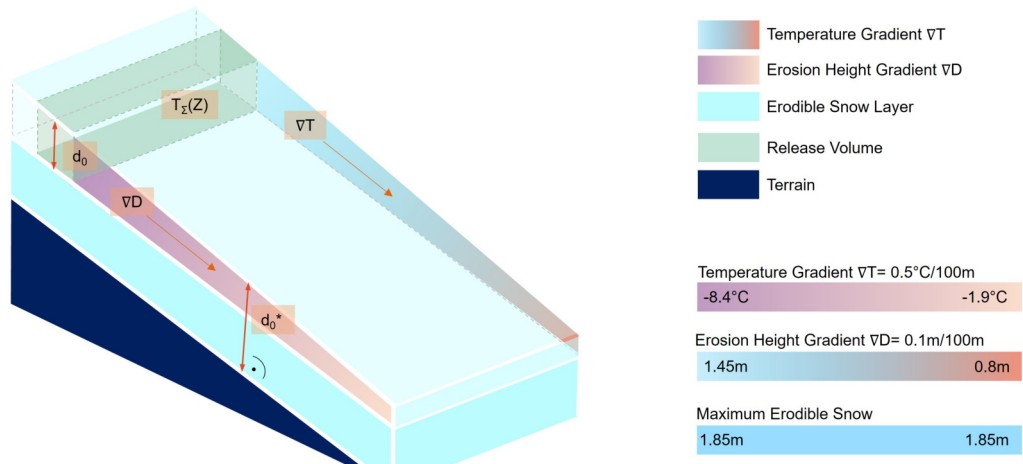

**Figure 6.** Graphical depiction of input values to initialize a simulation. On the right side of the graphic, an example for the values used for reproducing the Rüchi path is shown. The values are chosen based on field measurements and weather stations.

The gradient $\nabla D$ is expressed in m per 100 m change in elevation, see Figure 6. Higher snow accumulation heights are found at higher elevations. Typical gradient values for Switzerland (European Alps) are 0.03 m/100 m $\leq \nabla d_0 \leq 0.05/100$ m (Salm et al., 1990).

The next iteration $(d_0)^2$ accounts for the slope angle of the release zone. The height $d_0^{(1)}$ is adjusted with the slope reduction factor $f(\psi)$ (Salm et al., 1990)

$$f(\psi) = \frac{0.291}{\sin(\psi) - 0.202\cos(\psi)} \tag{2}$$

to calculate avalanche fracture heights,

$$d_0 = (d_0)^2 = f(\psi)d_0^{(1)}. \tag{3}$$

The slope reduction formula is derived by treating the new snow layer as a Mohr-Coulomb continuum governed by cohesion (c $\approx$ 600 Pa) and internal friction angle $\tan(\phi) = 0.202$ with density $\rho_0 = 200$ kg/m$^3$ (Burkard and Salm, 1990). It is based on the assumption of

We adopt the same two-step procedure to derive the erodible snow depth for road safety calculations. Moreover, we take

$$d_\Sigma(Z) = f(\psi)\left[d_m + \nabla D\left(Z - Z_m\right)\right]. \tag{4}$$

where we now replace the fracture zone elevation $Z_0$ with the slope elevation $Z$. This procedure thus places less snow on very steep track segments, for example on cliff-faces. In the following we do not take the guideline values for accumulation gradients, but the values derived directly from the measurements $\nabla D = 0.1$ m / 100 m, see Table 1 as described in Section 2. The resulting snow distribution is shown in Figure 5.

We see strong temperature gradients present in the upper layer of the snowpack (i.e. recently accumulated snow) in both the release and runout zones (see in the Appendix Figure A1 for the snowpit measurements at nearby stations at the altitude of the release zone and deposition). The release temperature is defined as the average temperature of the snowpack from the fracture depth to the surface, excluding the snow surface temperature from the calculation. The measurements indicate that low snow temperatures ($T_m \approx$ -8.5°C) exist in the avalanche release zones ($Z$ = 2300 m.a.s.l.), and higher temperatures in the runout zone ($T_m \approx$ -1.9°C) at $Z$ = 1600 m.a.s.l.. The temperature gradient $\nabla T$ = 0.5 °C/100m (Table 1 and Figure 5) can be determined from

$$T_\Sigma(Z) = T_m - \nabla T (Z - Z_m). \tag{5}$$

## 3 Avalanche Model

To back-calculate the observed avalanches, we utilise the enhanced version of the depth-averaged RAMMS model (Christen et al., 2010). The extended model encompasses the avalanche core (denoted by the Greek letter $\Phi$), the powder cloud (designated as $\Pi$), and the underlying snowcover ($\Sigma$), see Figure 7. The basics of the model are presented in (Zhuang et al., 2023a). In this chapter, we will revisit some of the key equations from that publication to provide a complete overview. Along with these equations, we will include more detailed explanations and introduce the closure relations.

To accurately model the observed avalanches and snowcover conditions, the following model features are necessary and contained in the extended RAMMS model: (1) Computation of the mean internal energy (thermal temperature) of the avalanche core given the initial temperature of the snowcover. (2) The ability to define snow cover properties as a function of elevation, exposition and slope-dependent terrain features. (3) Track the generation and independent propagation of the powder cloud. Development of the model has been conducted incrementally by Bartelt and Buser, and their collaborators (Bartelt et al., 2006; Buser and Bartelt, 2009; Bartelt et al., 2012, 2015a; Zhuang et al., 2023b). A first 1D two-layer model for powder snow avalanches including entrainment was developed at the beginning of the 1980s by Eglit (1982). Further models were developed by Russian researcher (Bozhinskiy and Losev, 1998); however, these models did not include grain flow process physics (Haff, 1983; Hutter et al., 1987; Jenkins and Mancini, 1987) or thermal effects (Valero et al., 2015, 2018). In the following sections, we present the model equations for the core, the cloud and discuss the role of the incumbent snowcover. The equations are based on the mass flow between the snowcover, core, cloud and air. As we calculate with the mass divided by the density and unit footprint area, we will show the equations as a function of height $H$. The mass flux which is also normalized over density and unit area across the boundary between two layers will be denoted by $\dot{H}$.

### 3.1 Avalanche core $\Phi$

The avalanche core is a shear flow containing mass in the form of snow clods (grain flow). The core dynamics are characterised by three state variables: namely, the co-volume height $\hat{h}_\Phi$, representing the snow packing found in the deposition zone, the dispersed or flowing height $h_\Phi$, and the slope-parallel velocity vector $\boldsymbol{u}_\Phi$. The co-volume height has the associated density $\hat{\rho}_\Phi$, whereas the dispersed flow height has density $\rho_\Phi$. The model assumes constant density and velocity profiles where the

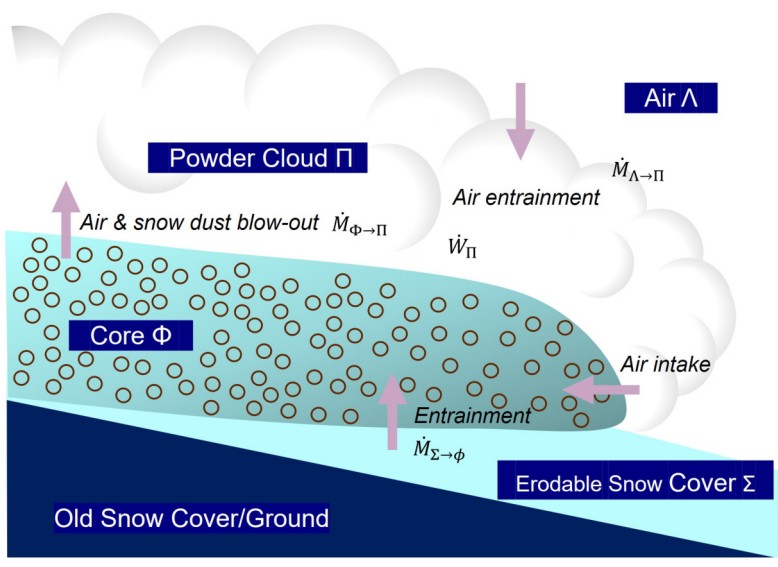

**Figure 7.** The three primary components of the extended RAMMS model are the avalanche core $\Phi$, the powder cloud $\Pi$ and the erodible snowcover $\Sigma$. The surrounding air is denoted $\Lambda$.

mean values are taken. Hence, to estimate damage areas, the profile function must be post-processed by the user. The mass and momentum equations for the avalanche core $\Phi$ are,

$$\partial_t \hat{h}_\Phi + \boldsymbol{\nabla} \cdot (\hat{h}_\Phi \boldsymbol{u}_\Phi) = \frac{\rho_\Sigma}{\hat{\rho}_\Phi} \dot{H}_{\Sigma \to \Phi} - \dot{H}_{\Phi \to \Psi} - \frac{\hat{\rho}_\Pi}{\hat{\rho}_\Phi} \dot{H}_{\Phi \to \Pi} \tag{6}$$

$$\partial_t h_\Phi + \boldsymbol{\nabla} \cdot (h_\Phi \boldsymbol{u}_\Phi) = \mathcal{D}(x,y,z) \tag{7}$$

$$\partial_t (\hat{h}_\Phi \boldsymbol{u}_\Phi) + \boldsymbol{\nabla} \cdot \left( \hat{h}_\Phi \boldsymbol{u}_\Phi \otimes \boldsymbol{u}_\Phi + p_\Phi I \right) = \boldsymbol{G} \hat{h}_\Phi - \frac{\boldsymbol{u}_\Phi}{||\boldsymbol{u}_\Phi||} \boldsymbol{S}_\Phi - \left[ \dot{H}_{\Phi \to \Psi} + \frac{\hat{\rho}_\Pi}{\hat{\rho}_\Phi} \dot{H}_{\Phi \to \Pi} \right] \boldsymbol{u}_\Phi. \tag{8}$$

The avalanche core is driven by gravity $\boldsymbol{G}$ and resisted by shear stress per unit density $\boldsymbol{S}_\Phi$. The mass and momentum balances involve the snowcover entrainment $\dot{H}_{\Sigma \to \Phi}$, snow detrainment by trees $\dot{H}_{\Phi \to \Psi}$, and mass/momentum transfer to the cloud $\dot{H}_{\Phi \to \Pi}$. As a simplification, the mass exchange between the dense core and the suspension layer is considered unidirectional, 215  with no mass from the cloud being drawn back into the core. The implications of this assumption are discussed in (Issler et al., 2024). Parametrization of forest detrainment is discussed in detail in (Feistl, 2015). Eq.7 describes the dilution and compression of the core; $\mathcal{D}(t)$ represents the change in core height due to dispersive pressure effects (Buser and Bartelt, 2015).

We calculate volumes of air that enter the core as the core expands and contracts. Hence the equation tracks the centre-of-mass of the granular ensemble. The equation is both a conservation (of air) and evolution (centre-of-mass) equation. A critical discussion on the approach of modelling the acceleration of the bed-normal expansion of the core's centre of mass directly on the bed-normal component of gravity can be found in Issler et al. (2017). $\mathcal{D}(t)$ is found by linking the mass and momentum equations to a grain flow equation for the fluctuation energy (granular temperature) $R_\Phi$, see (Haff, 1983; Jenkins and Savage, 1983; Hutter et al., 1987; Jenkins and Mancini, 1987; Buser and Bartelt, 2009) (for more details see Appendix.2),

$$\partial_t(\hat{h}_\Phi R_\Phi) + \boldsymbol{\nabla} \cdot (\hat{h}_\Phi R_\Phi \boldsymbol{u}_\Phi) = \alpha_\Phi \dot{W}_\Phi - \dot{H}_{\Phi \to \Pi} R_\Phi - \beta_\Phi \hat{h}_\Phi R_\Phi + \epsilon_\Phi \rho_\Sigma \dot{L}_{\Sigma \to \Phi}. \tag{9}$$

The fluctuation energy $R_\Phi$ is associated with random and dispersive particle movements in the flowing granular ensemble. It is produced by shearing $\dot{W}_\Phi$ (parameter $\alpha_\Phi$) and decaying by collisions/rubbing (parameter $\beta_\Phi$), see (Haff, 1983; Jenkins and Savage, 1983; Bartelt et al., 2006). These parameters are temperature-dependent. Due to limited data between cold and warm avalanches, a constant alpha of 0.07 is assumed for cold avalanches and 0.05 for warm avalanches. The calibrated curve for $\beta$ is shown in Appendix.2. Random particle movements are likewise produced during the entrainment process at the rate $L_{\Sigma \to \Phi} = 1/2 \dot{H}_\Sigma u_\Phi^2$ (parameter $\epsilon_\Phi$), see (Bartelt et al., 2018). Moreover, snow cannot be entrained in the avalanche without disrupting the mean flow. The counterpart to the macroscopic random fluctuations is identified as another form of stochastic energy, denoted as the internal energy $E_\Phi$, which is the complementary part of the macroscopic random fluctuations

$$\partial_t(\hat{h}_\Phi E_\Phi) + \boldsymbol{\nabla} \cdot (\hat{h}_\Phi E_\Phi \boldsymbol{u}_\Phi) = [1 - \alpha_\Phi] \dot{W}_\Phi - \dot{H}_{\Phi \to \Pi} E_\Phi + \beta_\Phi \hat{h}_\Phi R_\Phi + [1 - \epsilon_\Phi] \rho_\Sigma \dot{L}_{\Sigma \to \Phi} + \rho_\Sigma c_\Sigma T_\Sigma \dot{H}_{\Sigma \to \Phi} - \dot{Q}_m - q_{\Phi \to \Lambda}. \tag{10}$$

The model therefore predicts the mean avalanche temperature $T_\Phi$ which is related to the internal energy $E_\Phi = \hat{\rho}_\Phi c_\Phi T_\Phi$, where $c_\Phi$ is the specific heat capacity of snow at the density $\rho_\Phi$. We assume frictional heating processes due to shearing and rubbing between the particles. The term $\dot{Q}_m$ is a heat flux representing the energy used to melt snow,

$$\int_0^{dt} \dot{Q}_m dt = \rho_\Phi c_\Phi \hat{h}_\Phi [T_\Phi - T_m]. \tag{11}$$

The constant $T_m$ is the melting temperature of ice. The associated change in mass and energy loss due to the melting process is $\dot{Q}_m/L$ where $L$ is the latent heat of ice. Hence, the integral describes that all specific heat energy above the melting is available to drive latent heat exchanges during the time interval $dt$. The remaining terms on the right-hand side account for the addition of heat energy from entrained snow and the fraction of heat energy produced during the plastic collision of the snowcover and finally the sensible heat exchange ($q_{\Phi \to \Lambda}$) of the flowing snow with the air. An additional mass balance equation accounts for the intake of bonded water in the snowcover and melting (Valero et al., 2018)

$$\partial_t H_\Phi + \boldsymbol{\nabla} \cdot (H_\Phi \boldsymbol{u}_\Phi) = \frac{\rho_\Sigma}{\rho_w} \eta_w \dot{H}_{\Sigma \to \Phi} + \frac{\dot{Q}_m}{\rho_w L}. \tag{12}$$

The parameter $\eta_w$ defines the volumetric water fraction in the entrained snow.

## 3.2 Flow friction

The inclusion of the state variables ($R_\Phi$, $T_\Phi$) allows us to define a process based frictional resistance for the avalanche core $\Phi$ which is governed by material constants (Table 2). We apply a modified Voellmy-type friction law for flowing snow,

$$S_\Phi(R_\Phi) = \mu_\Phi R_\Phi N_\Phi + (1 - \mu_\Phi R_\Phi) N_0 \left[ 1 - \exp\left( -\frac{N_\Phi}{N_0} \right) \right] + \rho_\Phi g \frac{||\boldsymbol{u}_\Phi||^2}{\xi_\Phi R_\Phi}, \tag{13}$$

where $N_\Phi$ is the basal normal stress; $\mu_\Phi(R_\Phi)$ Coulomb friction; $\xi_\Phi(R_\Phi)$ velocity dependent friction and $N_0$ the so-called cohesion (Bartelt et al., 2015b). This empirical formulation was calibrated based on chut experiments with flowing snow (Platzer et al., 2007b, a; Bartelt et al., 2015b). When $N_0 = 0$, the formula collapses to the traditional Voellmy relationship (Salm, 1993). The formula therefore allows us to exploit, if necessary, the long historical knowledge and well-calibrated sets of Voellmy parameters used by practitioners, see (Salm et al., 1990).

The same experiments with flowing snow reveal a strong frictional hysteresis between the front and tail of the flow, indicating a process, or flow-dependent relation (Platzer et al., 2007b). Avalanche flow structure, now readily observed in field experiments (Sovilla et al., 2008), is likewise controlled by the frictional hysteresis between front and tail. Moreover, flow resistance at the front of the avalanche differs from the friction at the avalanche tail (Bartelt et al., 2007, 2012). This has significance for the determination of the frictional constants. We note that when $R_\Phi = 0$, we have the co-volume, or non-dispersive (dense, plug,
tail) friction values $\mu_0 = \mu_\Phi(R_\Phi = 0)$ and $\xi_0 = \xi_\Phi(R_\Phi = 0)$. Lower friction values at the avalanche front are found via

$$\mu_\Phi(R_\Phi) = \mu_0 \exp\left[ -\frac{R_\Phi}{A_\Phi} \right] \qquad\qquad \xi_\Phi(R_\Phi) = \xi_0 \exp\left[ \frac{R_\Phi}{A_\Phi} \right] \tag{14}$$

where $A_\Phi$ is the so-called activation energy (Bartelt et al., 2012). In this model, the avalanche front dynamics, responsible for the formation of the powder cloud, is mathematically represented as the region of the avalanche with higher fluctuation energies $R_\Phi$. The parameters ($\mu_0$, $\xi_0$, $A_\Phi$) are found via experiments, but more importantly, by back-calculation of measured
avalanche deposits (Bartelt et al., 2012). The spatial distribution of avalanche deposits in the field provides the additional needed information to determine friction parameters. For example, they can be immediately estimated in field visits by noting the steepest slope $\psi$ with avalanche snow $\tan(\psi) \approx \mu_0$, as deposition begins when $R_\Phi \to 0$. The location of the frontal deposits (runout) and the terminal velocity of the avalanche are necessary to calibrate the $\xi_0$ and the activation energy $A_\Phi$. Table 2 lists the recommended frictional values we take for avalanching after three-day new snowfall accumulations.

For our present purposes to investigate cold, mixed flowing avalanches appearing after new snowfall periods, we will take the model parameters ($\mu_0$, $\xi_0$, $N_0$, $A_\Phi$, $\alpha_\Phi$) to be temperature independent constants (Table 2). The only temperature dependent parameter will be the decay of fluctuation energy $\beta_\Phi(T_\Phi)$ with $T_\Phi$ given in °C. With the arctangent relationship we model it as

$$\beta_\Phi(T_\Phi) = 1.40 + \frac{1.6}{\pi} \arctan(1.6(T_\Phi - 271.5)) \tag{15}$$

to ensure that the decay coefficient is within the range $0.6/\,\mathrm{s} \leq \beta_\Phi \leq 2.0/\,\mathrm{s}$. As the inverse of $\beta_\Phi$ physically represents the lifetime of the fluctuation energy $R_\Phi$ it is linked to the onset of deposition and the flow structure of the avalanche (formation

**Table 2.** Overview of the fixed parameters in RAMMS::EXTENDED with the corresponding publications where the calibrations are described in more detail.

| Model parameter | Definition | Constant values 3-day snowfall | Thermodynamic constraint | Comment How to determine |
|---|---|---|---|---|
| $\mu_0$ | Coulomb friction | $\mu_0 = 0.55$ | $\mu_0 > 0$ | Controls runout<br>Onset deposition<br>Chute experiments<br>Field observations<br>(Platzer et al., 2007b) |
| $\xi_0$ | Velocity-squared friction (m/s$^2$) | $\xi_0 = 1800$ | $\xi_0 > 0$ | Controls velocity<br>Field experiments<br>(Bartelt et al., 2012)<br>(Zhuang et al., 2023b) |
| $N_0$ | Cohesion (Pa) | $N_0 = 200$ | $N_0 \geq 0$ | Controls runout<br>Chute experiments<br>(Bartelt et al., 2015b) |
| $A_\Phi$ | Activation energy (kJ) | $A_\Phi = 2$ | $A_\Phi > 0$ | Controls spatial distribution<br>of avalanche deposits<br>(Bartelt et al., 2012) |
| $\alpha_\Phi$ | Generation $R_\Phi$ (-) | $\alpha_\Phi = 0.07$ | $0 \leq \alpha_\Phi \leq 1$ | Controls flow density (front)<br>Controls avalanche length<br>Powder cloud formation<br>Powder cloud height<br>(Dreier et al., 2016)<br>(Zhuang et al., 2023b) |
| $\beta_\Phi$ | Decay $R_\Phi$ (-) | $\beta_\Phi(T_\Phi)$<br>Eq. 15 | $\beta_\Phi > 0$ | Controls flow density (front)<br>Controls avalanche structure<br>Tail formation<br>Controls spatial distribution<br>of avalanche deposits<br>(Bartelt et al., 2012) |

of the avalanche tail). It can therefore be determined by measuring the distribution of deposits in the avalanche runout zone (Bartelt et al., 2012). The lifetime of the fluctuation energy decreases as the avalanche temperature increases; it is approximately four times longer in a cold avalanche than a warm avalanche. This ensures that warm, moist avalanches have plug-like flow

regimes (Köhler et al., 2018).

Finally, we presently do not consider the influence of generated meltwater on the frictional constants, assuming that the snow temperature remains below the melting temperature of ice.

## 3.3 Powder cloud $\Pi$

A comparable set of partial differential equations is proposed to model the powder cloud $\Pi$. The air-blast is simulated by equations governing mass (Eqs.16 and 17) and momentum balance (Eq. 18), along with supplementary equations related to the generation and dissipation of turbulent fluctuations (Eq.19):

$$\partial_t \hat{h}_\Pi + \boldsymbol{\nabla} \cdot (\hat{h}_\Pi \boldsymbol{u}_\Pi) = \dot{H}_{\Phi \to \Pi} \tag{16}$$

$$\partial_t h_\Pi + \boldsymbol{\nabla} \cdot (h_\Pi \boldsymbol{u}_\Pi) = \dot{H}_{\Lambda \to \Pi} + \frac{\rho_i - \hat{\rho}_\Pi}{\rho_i - \rho_\Lambda} \dot{H}_{\Phi \to \Pi}. \tag{17}$$

$$\partial_t (\hat{h}_\Pi \boldsymbol{u}_\Pi) + \boldsymbol{\nabla} \cdot \left( \hat{h}_\Pi \boldsymbol{u}_\Pi \otimes \boldsymbol{u}_\Pi + p_\Pi \mathbb{I} \right) = \frac{\hat{\rho}_\Pi - \rho_\Lambda}{\hat{\rho}_\Pi} \boldsymbol{G} \hat{h}_\Pi + \dot{H}_{\Phi \to \Pi} \boldsymbol{u}_\Phi - \frac{\boldsymbol{u}_\Pi}{||\boldsymbol{u}_\Pi||} \boldsymbol{S}_\Pi - \frac{\rho_\Lambda}{\hat{\rho}_\Pi} \dot{H}_{\Lambda \to \Pi} \boldsymbol{u}_\Lambda. \tag{18}$$

$$\partial_t (\hat{h}_\Pi R_\Pi) + \boldsymbol{\nabla} \cdot (\hat{h}_\Pi R_\Pi \boldsymbol{u}_\Pi) = \dot{W}_\Pi + \dot{H}_{\Phi \to \Pi} R_\Phi + \frac{1}{2} \rho_\Lambda \dot{H}_{\Lambda \to \Pi} ||\boldsymbol{u}_\Pi||^2 - \beta_\Pi \hat{h}_\Pi R_\Pi. \tag{19}$$

The initial cloud height is denoted as $\hat{h}_\Pi$ with the corresponding initial cloud density $\hat{\rho}_\Pi$, representing the cloud state before the expulsion from the core. The variable $h_\Pi$ denotes the actual cloud height which results from dust-air mixture expelled from the core $\dot{H}_{\Phi \to \Pi}$ and air entrainment $\dot{H}_{\Lambda \to \Pi}$. The cloud density decreases due to air entrainment denoted by $\rho_\Pi$ . The actual cloud density must fulfil the following relation ship $\rho_\Pi = \rho_i \frac{\phi_i \hat{h}_\Pi}{h_\Pi + \phi_i \hat{h}_\Pi} + \rho_\Lambda \frac{h_\Pi}{h_\Pi + \phi_i \hat{h}_\Pi}$, where $\rho_i = 917$ kg/m$^3$ is the ice density, $\rho_\Lambda = 1.225$kg/m$^3$ is the air density, and $\phi_i = \frac{\hat{\rho}_\Pi - \rho_\Lambda}{\rho_i - \rho_\Lambda}$ represents the ice fraction in the initial cloud. The density of air varies slightly with changes in temperature, pressure, and humidity, but this variation has a negligible effect on the simulation. Therefore, we have chosen to keep the value fixed. The cloud is propelled by the momentum imparted from the core $\dot{H}_{\Phi \to \Pi} \boldsymbol{u}_\Phi$ and gravity $\frac{\rho_\Pi - \rho_\Lambda}{\rho_\Pi} \boldsymbol{G}_\Pi$. Generally, we observe $\dot{H}_{\Phi \to \Pi} \boldsymbol{u}_\Phi \gg \frac{\rho_\Pi - \rho_\Lambda}{\rho_\Pi} \boldsymbol{G}_\Pi$. This observation remains valid as long as the dense core and the suspension layer move together. However, once the core comes to a stop and the cloud ascends a counter slope, the gravitational term will once again become more significant, as discussed in (Issler et al., 2017). Additionally, it is important to note that the mass flow between the cloud and core is assumed to be unidirectional. Consequently, the vertical expansion of the core, driven by increased granular temperature, prevents the formation of a vacuum within the core. As a result, a portion of the air-snow mixture from the cloud is inevitably drawn back into the core (Issler et al., 2024).

The turbulence in the cloud due to the fluctuation energy are described by equation 19. The structure of the equation is based on earlier one-layer 3D models (Hermann et al., 1994; Gauer, 1995; Sampl and Zwinger, 2004). The fluctuation energy is produced by three sources (Eq. 19): internal shearing $W_\Pi = [\hat{\rho}_\Pi S_\Pi] ||\boldsymbol{u}_\Pi||$, fluctuation energy transferred from the core

$\dot{H}_{\Phi \to \Pi} R_\Phi$ and air entrainment $\frac{1}{2}\rho_\Lambda \dot{H}_{\Lambda \to \Pi} u_\Pi^2$. $\beta_\Pi$ is the parameter that controls the decay of turbulence, and therefore the lifetime, of the fluctuation energy $-\beta_\Pi \hat{h}_\Pi R_\Pi$. The pressure $p_\Pi$ includes both the hydrostatic and turbulent parts. More details of the powder model equations, including the entrainment function $\dot{H}_{\Lambda \to \Pi}$, friction $\boldsymbol{S}_\Pi$ and turbulence parameters is contained in the publication (Zhuang et al., 2023b) and in the Appendix.2. The air entrainment function and friction is calibrated based on observed avalanches in VdlS and shown in the attachment Appendix.2.

In the cloud, we assume inelastic collisions between particles, such that all energy is converted into random motion, with no energy dissipated as heat, as a simplification. Hence, the constant $\epsilon_\Pi$ is set at 1.

### 3.4 Entrainment

Entrainment in the extended model equations is treated as a plastic collision between the avalanche core and the snow cover (Bartelt et al., 2018). We initially define the snow volume per unit footprint area and unit time which results in a height $\dot{H}_\Sigma$ that is affected by the passage of the avalanche core:

$$\dot{H}_\Sigma = \kappa_\Sigma \frac{\rho_\Sigma}{\hat{\rho}_\Phi} ||\boldsymbol{u}_\Phi||. \tag{20}$$

We represent this interaction rate as proportional to the avalanche speed $||\boldsymbol{u}_\Phi||$, as it determines the distance the avalanche travels during the interaction time. Avalanches moving at higher speeds cover more ground, leading to an increase in the amount of snow cover mass affected by the avalanche. In this model, even very thin avalanches could potentially erode the same amount of snow as a thick one (Issler et al., 2024). To prevent this, a shear stress cutoff is implemented, ensuring that an avalanche must exceed a certain energy threshold before it can erode snow.

We define the erodibility coefficient as the dimensionless parameter $\kappa$. Essentially, $\kappa$ determines the ratio of erosion speed of the erosion front in the snowcover in direction normal to the terrain relative to the flow velocity. Low $\kappa$ values indicate that only the surface of the snowcover is affected by the avalanche passage (basal erosion), while, conversely, high values of $\kappa$ indicate that the core affects the entire depth of the snowcover (frontal erosion). The value of $\kappa$ can be adjusted to incorporate a internal friction of the snow $\mu_b$. Defining $g_s$ to be the slope parallel acceleration and $g_z$ as the slope normal acceleration, we modify $\kappa$ to be

$$\kappa = \frac{\kappa'}{g}\left[g_s - \mu_b g_z\right] \quad \text{where} \quad \kappa \geq 0 \text{ always.} \tag{21}$$

For 3-day accumulation periods with new snow, we take $\kappa' = 0.015$ and $\mu_b \approx 0$. These values ensure that on steep slopes with high avalanche velocities we model frontal entrainment, whereas on more gentle slopes and at lower speeds the avalanche enters a mode of basal erosion. The bonding strength model is motivated by observations of eroded segments in avalanche tracks. On track segments where there are no depositions, it can be ascertained that erosion has occurred. In this case, the parameter $\mu_b$ must be smaller than the tangent of the slope angle.

Presently, $\dot{H}_\Sigma$ represents the snowcover mass affected by the avalanche core – not the total amount of snow taken in by the avalanche. We now partition the affected mass into two parts: a part of mass which is entrained by the avalanche, and a part of

the mass which is not entrained, possibly splashed in front of the core to build a pre-front, or frontal saltation layer $\Gamma$,

$$\dot{H}_\Sigma = \dot{H}_{\Sigma \to \Phi} + \dot{H}_{\Sigma \to \Gamma} \tag{22}$$

The mass flux $\dot{H}_{\Sigma \to \Phi}$ represents the snowcover mass that is accelerated to the avalanche velocity and can be found on the right-hand side of model equations Eq. 6 and Eq. 8. We apply a partitioning parameter $\gamma$ to separate the entrained/non-entrained fractions of the snowcover,

$$\dot{H}_{\Sigma \to \Gamma} = \gamma \dot{H}_\Sigma \qquad\qquad \dot{H}_{\Sigma \to \Phi} = (1 - \gamma) \dot{H}_\Sigma \tag{23}$$

The parameter $\gamma$, which we term the splashing parameter, could also represent the non-entrained mass in the disrupted snow-
cover that is simply accelerated by the passage of the avalanche front. Different snowcovers will be governed by different entrainment parameters ($\kappa'$, $\mu_b$ $\gamma$). For 3-day new snow accumulation periods we take for as a simplification $\gamma = 0.2$. For more moist avalanches, this value goes to zero.

## 4  Results and Discussion

In the following, we present the results of the sensitivity analysis on model input parameter. We delve deeper into the mathe-
matical model's representation of the effects of snow temperature and erosion (Section 4.1), release zone properties (Section 4.2) and friction parameters (Section 4.3). We conclude with a comparison to the measured avalanches (Section 4.4). All other system parameters were calibrated based on observations of avalanches in VdlS as presented in Chapter 3. As a base we used the model parameters shown in Table 2.

### 4.1  Snow Temperature and Erodible Snow Depth

In most avalanching situations, alpine snow is within a few degrees of its phase transition point ($T = 0$ °C). The physical properties of flowing snow undergo rapid transformations as temperatures edge towards the melting threshold. In the model equations the temperature dependence is contained in the decay parameter of granular temperature $\beta$ (Eq. 15) (Haff, 1983) which describes the decay of the random kinetic energy (granular temperature) and therefore the dispersion of the snow granules. The parameter is set such that colder avalanches exhibit the tendency to form mixed flowing avalanches (Bartelt
et al., 2012), while warmer avalanches will exhibit more plug-type flows (Li et al., 2021). By defining the avalanche release temperature, not only do we set the initial thermal energy, but we also dictate the predisposition of the avalanching snow towards dry, mixed flowing avalanche or moist flow regimes.

In a first series of numerical experiments, we applied the model on an idealized planar slope (Figure 1). The slope inclination was set to approximate the Brämabühl slopes under investigation. In this way secondary terrain features inducing flow channels
and secondary flow fingers could be removed from the analysis and model performance gauged in idealized conditions. We varied release temperatures from extremely cold temperatures to the melting point (-20 °C $\leq T_0 \leq$ 0 °C).

In the first simulations we included no entrainment $d_0^* = 0$, meaning, no additional snow was eroded by the avalanche. We calculated the runout distance according to our post-processing procedure of minimum heights and velocities (green dots,

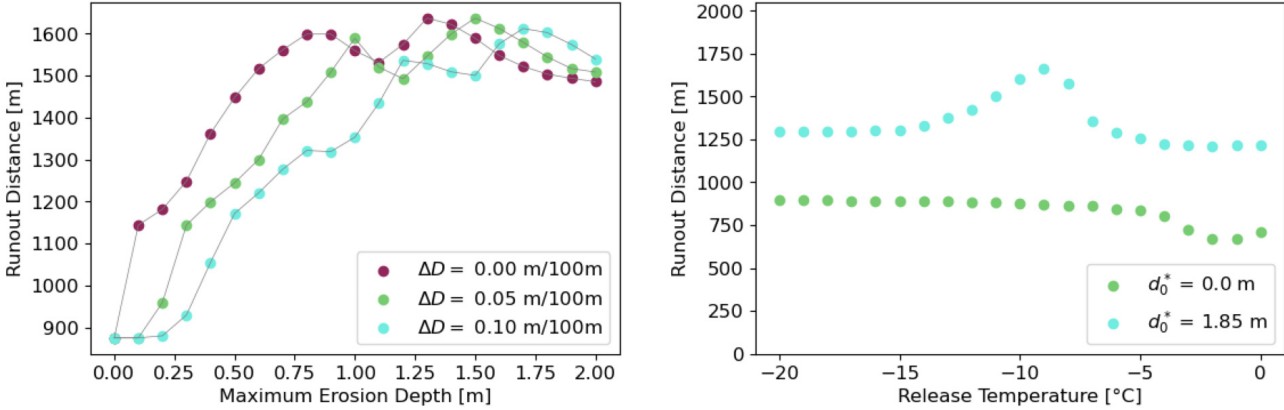

**Figure 8.** Simulation of runout distance of an avalanche on an inclined plane. On the left side: variation of maximum erosion depth for different erosion gradients and on the right side: variation of release temperatures. Simulation points for release temperatures near the melting point should be interpreted with caution, as the simulation tool is not yet well calibrated for this range.

Figure 8). In this case the runout distances remained constant over a wide temperature range (-20 °C $\leq T_0 \leq$ -8 °C). At higher temperatures the decay parameter of granular temperature increases, with the resulting effect of reduced fluidization and shorter runouts.

By incorporating erosion in our simulations, we add entrained mass to the flowing avalanches and their responses get more intricate. In a subsequent series of simulations we assume a deep snowcover $d_0^*$= 1.9m with a small height differentials $\nabla D$ = 0.1 m / 100 m and temperature gradients $\nabla T$ = 0.5 °C / 100 m (blue dots, Figure 8). At extremely cold temperatures -20 °C $\leq T_0 \leq$ -13 °C the calculated runout distances are independent of $T$. We assume the reason for this behaviour is the decay parameter of granular temperature $\beta$. For very cold avalanches, the formation of the powder cloud dominates, extracting mass and energy from the core, ultimately leading to dispersion and dissipation of the avalanche. As the temperatures increase, an optimal balance between the core and cloud emerges, yielding far-reaching flows. With the specification of more realistic release temperatures, $T > -13$ °C the runout distances increase. This phenomenon underscores the counteracting effects of frictional heating (rise in temperature) and the entrainment of cold snow (decrease in temperature). The avalanche temperature remains lower for longer, fostering long-lasting fluidized regimes and more potent powder avalanches. The model predicts that the entrainment of cold snow at lower elevations sustains the fluidized regime and the formation of powder avalanches. However, the runout distances decrease again with higher release and entrainment temperatures $T_0 > $-8°C (Figure 8). At higher temperatures the decay of fluctuation energy increases, leading to dense, less fluidized flows and therefore an increase in friction which curbs runout distances. It becomes evident that the temperature-dependent decay parameter of granular temperature, $\beta(T_\Phi)$, controls the flow regime of the simulated avalanches with and without entrainment.

In the next series of simulations on the idealized planar slope, we vary the snow height gradient $\nabla D$ = 0.0, 0.05 and 0.10 m/100 m; the release temperature remains set at $T_0$= $-6$°C and the temperature gradient is zero, $\nabla T$ = 0. Less snow is

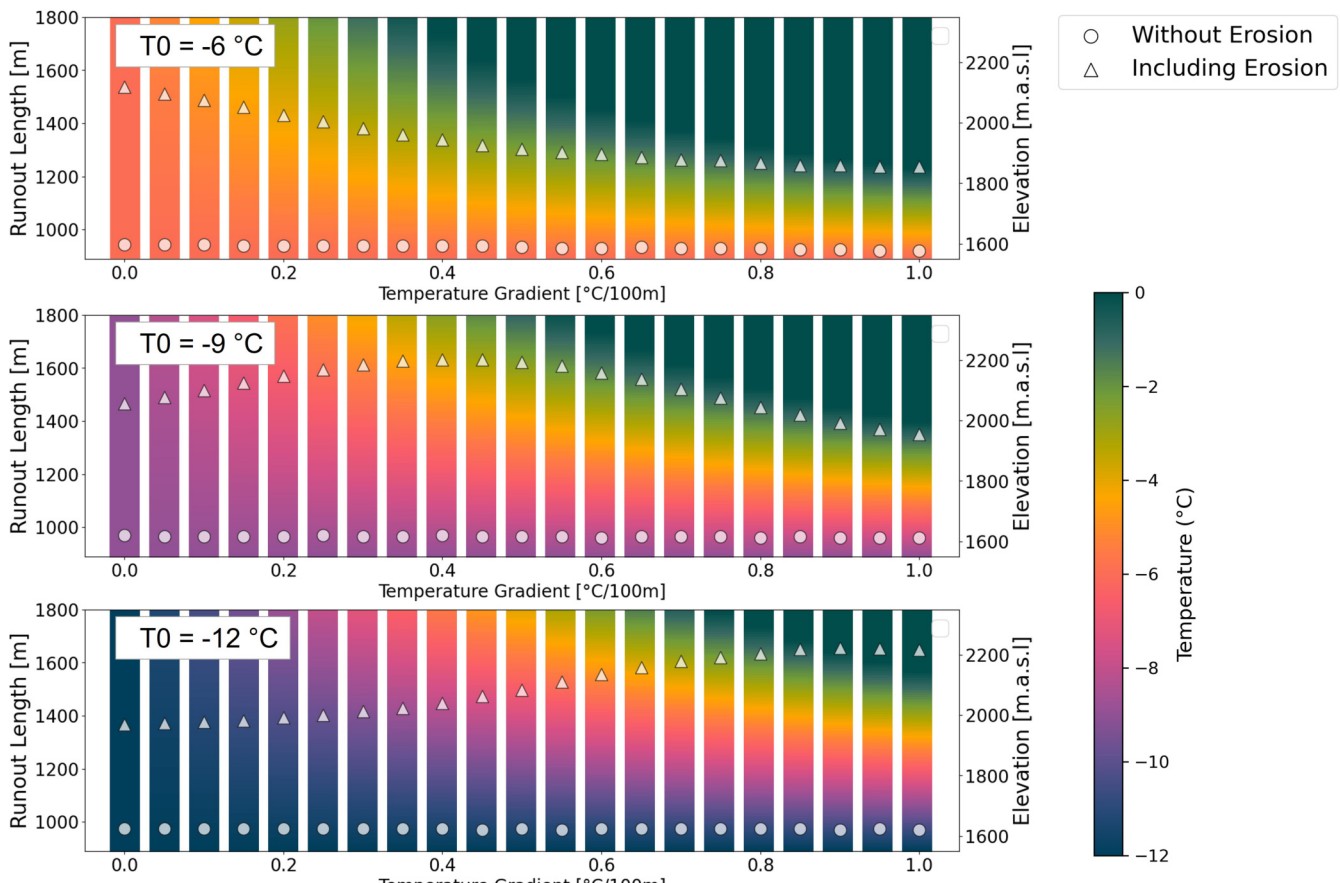

**Figure 9.** Simulation of avalanche runout on the idealized planar slope with and without erosion. We consider three initial temperatures (a) $T_0$ = -6 °C (b) $T_0$ =-9 °C and (c) $T_0$ = -9 °C for variable temperature gradients $\nabla T$. The colours on the blocks depict the snow temperature varying from cold (blue) to warm (green).

subsequently encountered by the avalanche at lower elevations and into the runout zone. We perform simulations with different
values of snow heights $0.0\,\text{m} \leq d_0^* \leq 2.0\,\text{m}$. This situation mirrors actual snow conditions in road safety applications in which the snow distribution could represents a large uncertainty. We simulate the effect of entrained snow of different height gradients for shallow and deep snowcovers. The results indicate that for shallow snowcovers $d_0^* \leq 1.0\,\text{m}$, the different gradients produced large differences in avalanche runout (Figure 8). Avalanches that encounter snow along the entire track ($\nabla D = 0.0\,\text{m}\,/\,100\,\text{m}$) run longer than those that encounter regions of no snow in the runout zone. This result corresponds well with the experience
that a deep snowcover from initiation to runout is needed for extreme avalanche events, especially powder snow avalanches. For the case of $d_0^* > 1.0\,\text{m}$, the gradients appear to have no influence on the avalanche runout; the snow is so deep that even a strong gradient still does not entirely remove the snowcover for this parameter set.

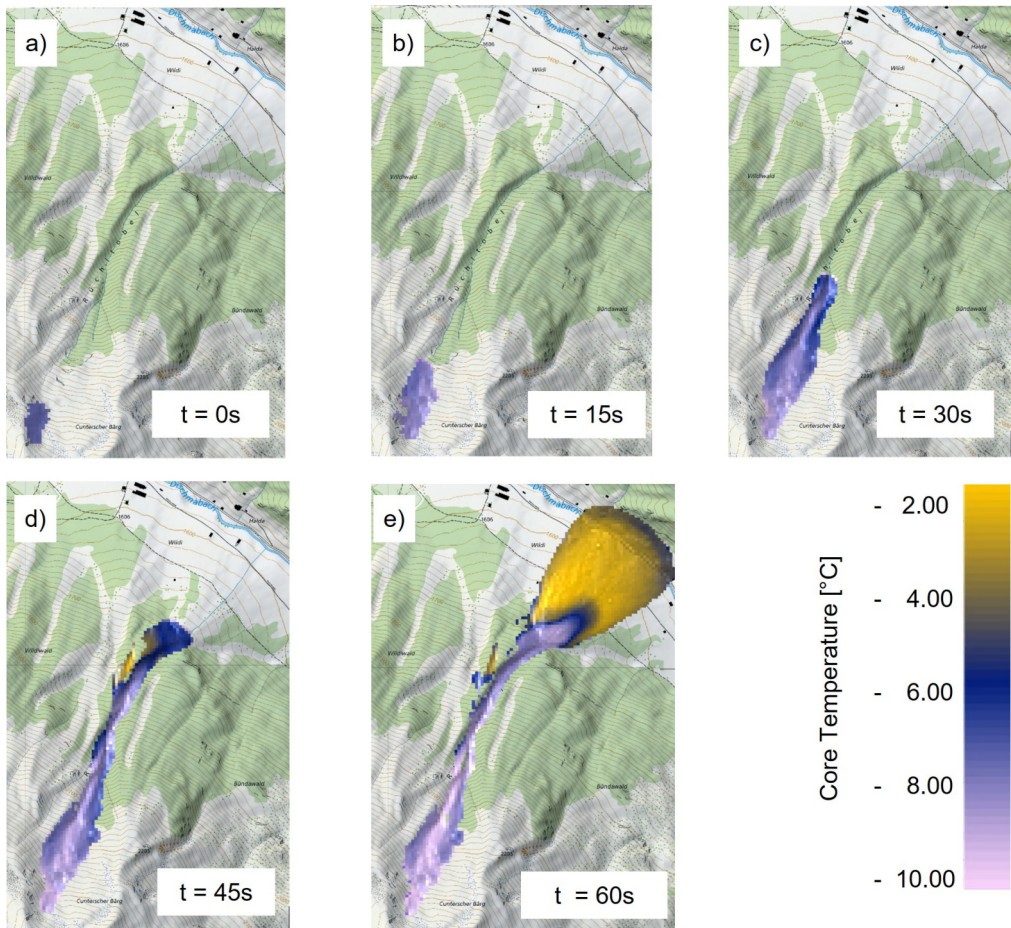

**Figure 10.** Temperature evolution in the Rüchi avalanche over time, $t = 0$ s (a) to $t = 60$ s (e) (map source: Federal Office of Topography).

Next, we investigated the effects of elevational snow temperature gradients in Figure 9. we show the decrease in avalanche runout for variable temperature gradients $\nabla T$ as we simulated it on the idealized planar slope. The colours in each panel depict the core temperature at various time steps throughout the avalanche flow. The analysis is done for different release temperatures. The results indicate the reduction in avalanche runout as a cold avalanche runs into a warm snowcover. The colder the initial temperature the less the reduction. The results underscore the complex interplay between initial release conditions, the process of entrainment, and the responding avalanche temperature.

The simulations on the idealized planar slope are useful because they highlight how the various snow parameters play an integral role for the avalanche outcome without localized terrain effects. On a actual slope the snowcover height is not only given by the elevation gradient $\nabla D$ but also the local slope inclination. The simulated evolution of avalanche temperature over time for the Rüchi path is depicted in Figure 10. The initial temperatures are specified using the weather station and snowpit data. This figure displays the calculated mean avalanche temperature $T_\Phi$ of the avalanche core, which increases by 3 °C from

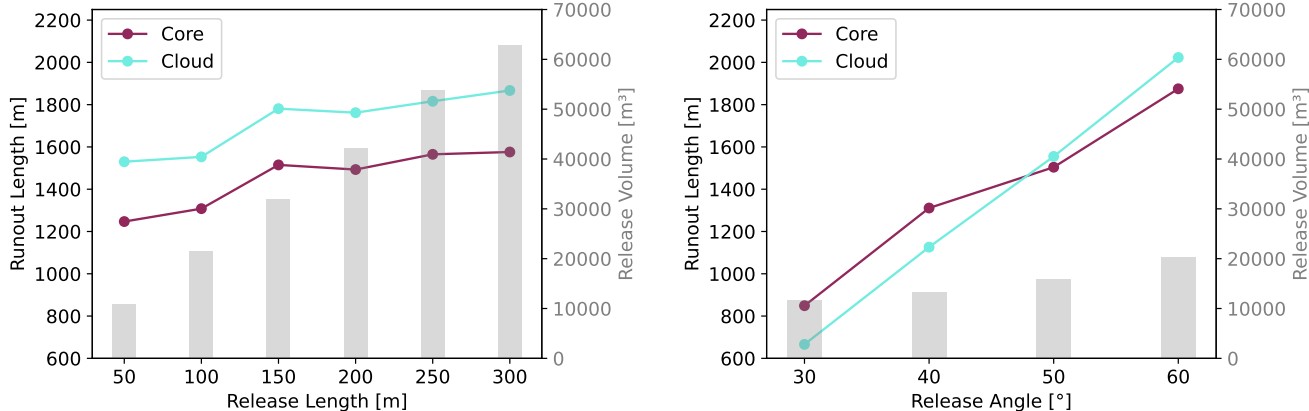

**Figure 11.** Representation of the avalanche runout length if the release length and the release angle are varied for the core and the cloud. In grey, the release volume is represented.

$T_\Phi$ = -8 °C to $T_\Phi$ = -2 °C in the runout zone. The calculated avalanche temperature results from the competition between frictional heating (avalanche velocity and therefore terrain) and intake of cold snow (Valero et al., 2015). The model equations assume that the entrained snow mixes with the avalanche snow instantaneously, producing a new mean temperature. In reality, the energy exchange between the avalanche snow and the entrained snow will happen fast but takes some time as shown in Köhler et al. (2018). Temperature variations will exist in the avalanche; heat concentrations will most likely exist on the surface of the granules, while the interior of the granules remains cold (Jomelli and Bertran, 2004; Steinkogler et al., 2015).

## 4.2 Release Zone

In this section, we investigate how the shape and location of the defined release zone influences avalanche simulations by varying the length and steepness of the release zone. We conduct our simulations on the idealized slope as shown in 4, but with an infinite runout of 28° slope to remove the influence of the flat runout zone. For all the simulations, we used the parameter set for the Rüchi avalanche as described in Table 1.

We used an idealized version of the Rüchi release zone which is a square with a width of 120 $m$ and a length in flow direction of 75 m. We varied the length up to 2.5 times the width, to stay in the ratio of avalanche width compared to length proposed by the Swiss guidlines (Hrsg., 2024) and the release shape proposed by (Mcclung, 2009). The release zones have an average release angle of 35°. As the release volume increases, RAMMS proposes a different volume-dependent friction parameter set for $\mu$ and $\xi$. As we want to analyse RAMMS with it's proposed calibration, we plot the values with the proposed friction values according to the release volume and do not hold them constant. By increasing the release length, we automatically increase the release volume and the avalanche starts with a higher potential energy (see Figure 11). The simulations show that we get longer avalanches with an increased release zone length.

For the variation in release angle, we used the same release zone throughout. In RAMMS::EXTENDED, the release zone is defined by its projected area, meaning that the actual release area becomes slightly larger as the release angle increases. Figure 11 shows that the runout distance increases almost linearly with an increase in release angle. Avalanches initiating at lower release angles show a powder cloud that is shorter in length compared to the core. Conversely, at release angles exceeding 48 ○, the powder cloud can achieve greater acceleration, flowing further than the core. On a real terrain, the terrain friction would strongly influence the avalanche core runout distance and we expect that this observation would change.

### 4.3 Friction Parameters $\mu_0$, $\xi_0$, $N_0$

In the preceding section all the calculations were performed with constant friction and process parameters (see Table 2) representing avalanche situations governed by periods of new snowfall. The simulation results indicate that given an initial release location, mass and temperature, the calculated terminal velocity and avalanche runout are governed by snowcover disposition and temperature. Traditionally, runout and velocity are reproduced in avalanche dynamics calculations by changing the values of the friction parameters from avalanche to avalanche creating an envelope of extreme values (Gruber and Bartelt, 2007). Here, we do not adopt this approach. Friction parameters dynamically change as a function of temperature according to the process chain,

$$T_\Phi(t) \to R_\Phi(t) \to \text{Voellmy parameters} \mu(t), \xi(t). \tag{24}$$

This chain of relationships indicates that the temperature of snow influences the mean fluctuation energy (via the decay parameter $\beta(T_\Phi)$), which controls the dispersion of snow granules and therefore avalanche flow regime. This fluctuation energy, being a stochastic variable, signifies the inherent randomness in the movement of all granules within the flowing snow ensemble relative to the mean. The momentary state of friction is influenced by this fluctuation energy. Thus, the basic model assumption is that the temperature of snow governs the stochastic dynamics of its granular ensemble, ultimately impacting frictional behaviour. The grain flow process parameters controlling the relationship between $R_\Phi(t)$ and flow friction (avalanche deposition) have been identified by Bartelt et al. (2012) in the analysis of experimental data from the test site Vallée de ls Sionne.

It is now necessary to validate this approach using the Brämabühl avalanches using in-situ information of snow distribution and weather station records. For this the initial values of the friction parameters $\mu_0$ and $\xi_0$ were varied to identify the combination which results in the measured avalanche runout distance. This was also done for different cohesion values. The results are depicted in Figure 12. For some cohesion values, the measured runout distance could not be reproduced. Therefore, we restricted the investigated cohesion values to the range obtained from measurements conducted in snow chutes (Bartelt et al., 2012).

We used the observed runout distance to determine the optimal friction parameters. We find that the best fit of all three Brämabühl avalanches runout distances is provided by friction values $0.55 \leq \mu_0 \leq 0.50$ and $1750 \text{ m/s}^2 \leq \xi_0 \leq 2200 \text{m/s}^2$. This is in good agreement with values found in VdlS calibrations (Bartelt et al., 2012). The range of $\xi_0$ could be reduced by knowing the avalanche core velocity. in Figure 12, the Rüchi path exhibits the most symmetric pattern in terms of friction values. This is attributed to the fact that the Rüchi path features a straightforward avalanche outline without any flow fingers.

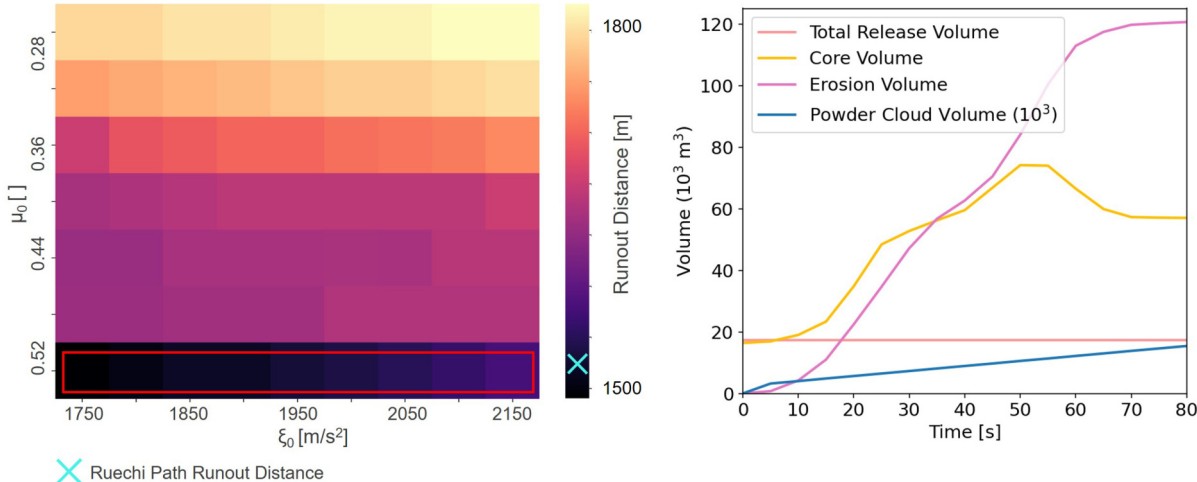

**Figure 12.** Right side: Simulation of the Rüchi avalanche with different friction values and a cohesion of $N_0$=150 Pa. The measured runout length of 1545 m $\pm 5\%$ error (over and underprediction of the avalanche runout) is marked by the red rectangle. The best-fit parameters are near the recommended values in Table 2 Left side: Evolution of avalanche volume along the Rüchi path from the simulation.

## 4.4 Comparison to Measured Avalanches

Photogrammetric data collected with the drone provided important information on the lateral flow width of the avalanche, the avalanche volume and the height and travel distance of the powder cloud.

Figure 13 depicts the calculated extent of the model avalanches in comparison to the measured outlines. The relatively good agreement between the calculated and measured avalanche flow widths is particularly significant.

Each avalanche was accompanied by a powder cloud that ran up the counter slope (Figure 14). The calculated powder cloud widths are in good agreement with the observations. By comparing the photographs of the fully developed powder cloud with the tree heights known from a LiDAR-based vegetation height model; and knowing that maximum air-blast pressures on the road never exceeded 5 kPa (as houses and windows remained undamaged), we estimate that the maximum powder cloud heights reached up to 40 m. Regions of isolated tree damage by the powder cloud are reproduced by the model. Additionally, comparing the estimated powder cloud velocities (Figure 2) for the Rüchi and Chaiseren path in the runout zones shows that they are in the same range as the maximum velocity per pixel resulting from the simulation as shown in Figure 14.

in Figure 15, we compare the snow height in the deposition zone of the Rüchi path to the simulated deposition height. The map of the measured snow height shows the evolution of deposited avalanche arms. Particularly on the right side in the flow direction of the avalanche, a stronger arm has formed, which is also evident in the simulation. Photogrammetric snow depth mapping with the drone revealed that the total snow volume present in the area overflowed by the avalanche in Figure 15 was approximately 155000 m$^3$. The simulated deposition in the same area was only approximately 30000 m$^3$. The drone measurements assess the total volume of snow present at acquisition time. As the avalanche ran on an an already present snow

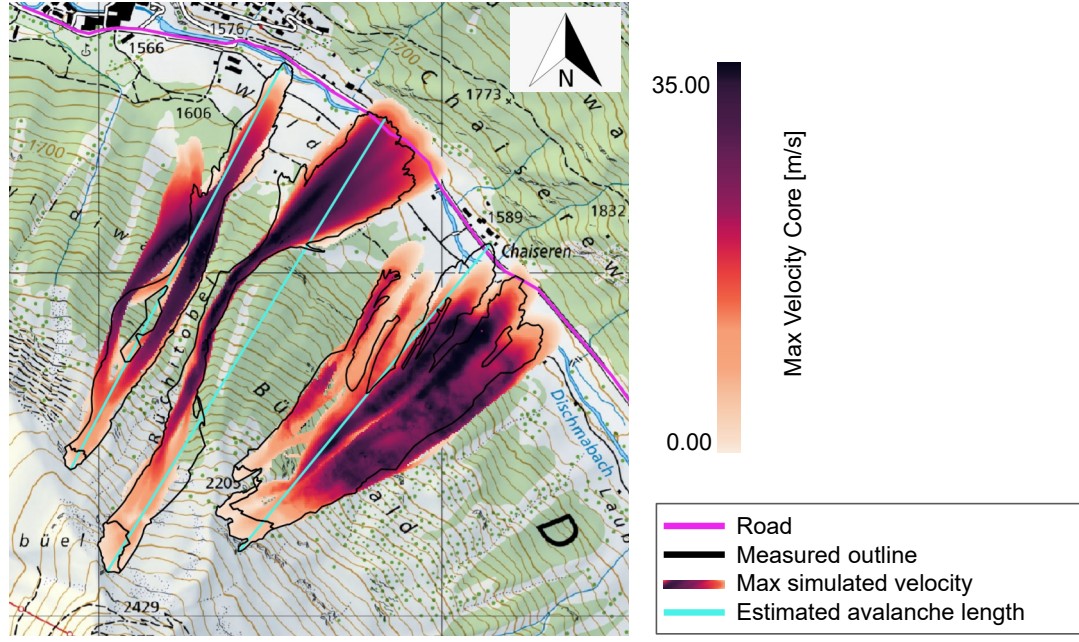

**Figure 13.** Comparison of the measured outlines of the Brämabühl avalanches with the model results using the measured snowcover height and temperature data (map source: Federal Office of Topography).

cover of approximately 1 m (measured next to the avalanche deposit, resulting in a already present volume of approximately
100000 m³), this is the main explanation for the difference. Furthermore, a large portion of the deposition in the simulated avalanche is already deposited further up in the avalanche track and the front stopped just outside of the mapped avalanche outline and is therefore not taken into account for the volume calculation. The depositions simulated in the upper part of the track are also present in the drone measurements and the simulations at similar locations Figure 13 and show similar deposition heights.

The drone orthophoto of the Chaiseren avalanche track showed that its snowpack had been scoured by wind, resulting in less accumulated snow than was calculated by the snow gradients used in our simulations. Therefore, the initial snow mass is over-estimated in this simulation.

## 5   Conclusion

For road safety managers it is essential to know, based on near real-time weather and snowpack information, whether an
avalanche could reach the road and a road closure is warranted. With the goal to better support their decision-making process, we tested an approach where we adopted a numerical model (RAMMS::EXTENDED) to investigate the sensitivity of various snow parameters, that can be measured in the field, such as, release zone location and length, release temperature, and gradients for snow temperature and snow height in the path, and the avalanche response to changes in these parameters. The study

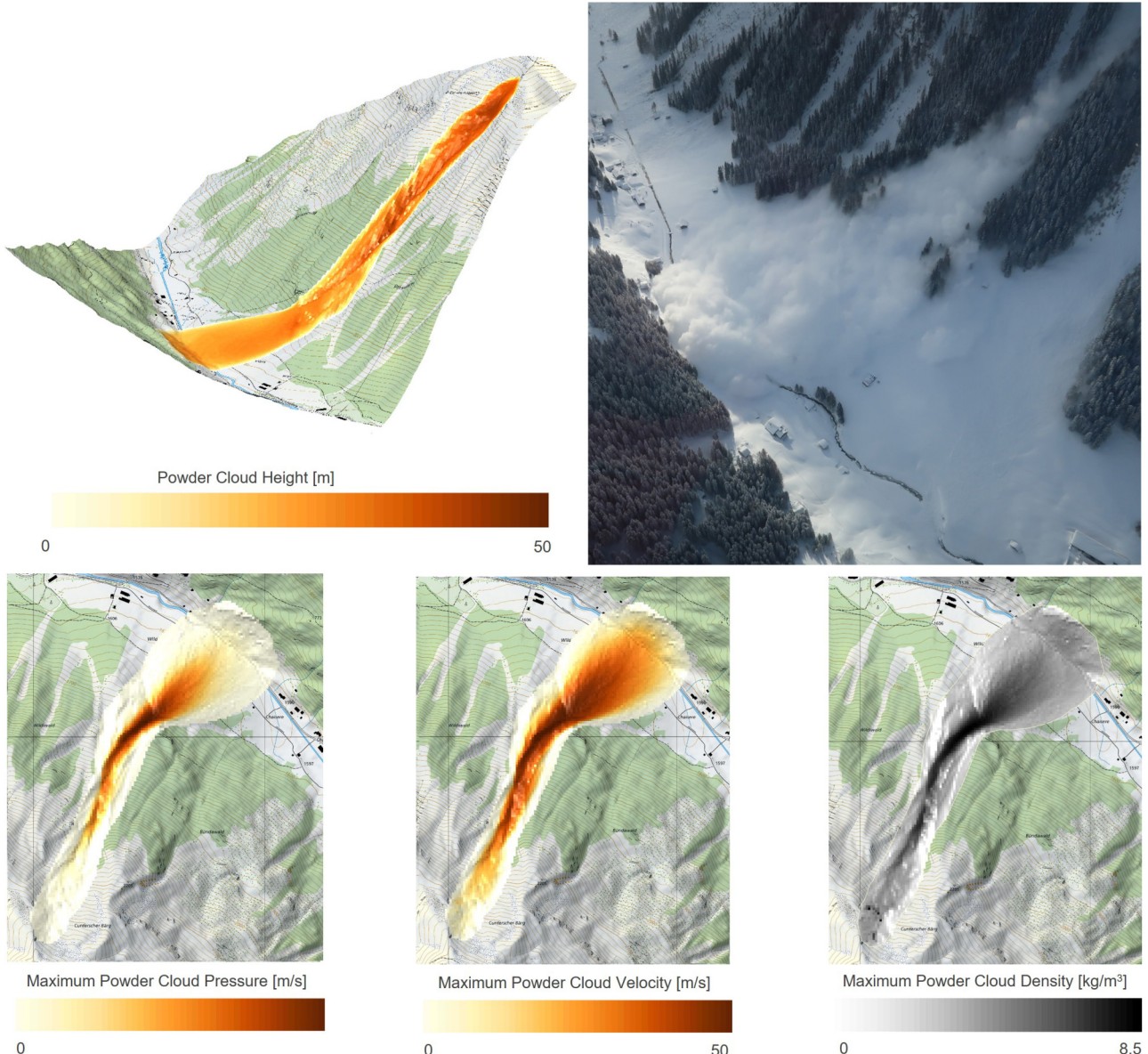

**Figure 14.** Photo of the cloud taken from the helicopter of the Rüchi avalanche and the simulation results from the back-calculation showing the maximum cloud height, pressure, velocity and cloud density. All values shown are maximum values per pixel (map source: Federal Office of Topography).

presented herein was based on an avalanche cycle from the Dischma Valley, Switzerland, in 2019 that caused several avalanches to cover the road during a post-storm cold spell. These avalanches were well-documented and could therefore be used for avalanche back-calculations. For the purpose of this study, we only simulated cold powder avalanches; both the core and the

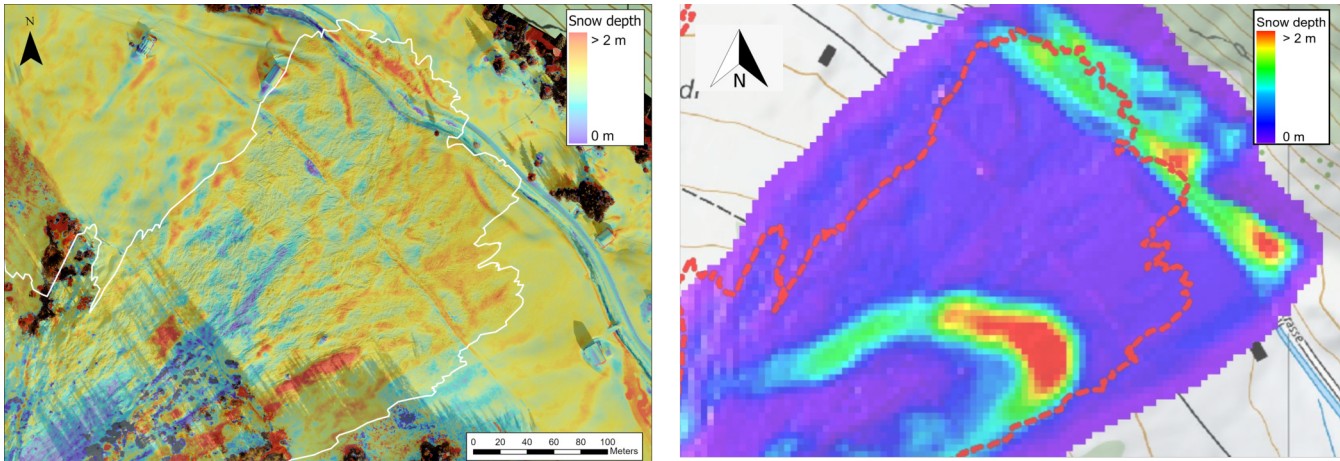

**Figure 15.** Comparison of the deposition area of the core measured from the drone data (left) and modelled (right) Rüchi avalanche. The grid lines mark the distance of 1 km (map source: Federal Office of Topography).

powder cloud of these mixed avalanches could pose a hazard to cars and people. The sensitivity analysis was mainly based on simulations on an idealized plane, resembling the topographic profile of the Ruchi avalanche path. The influence of terrain effects, e.g. on friction parameters in the model, was investigated at a second phase of the analysis where we reconstructed the avalanche events in Wildi, Rüchi and Chaiseren paths that took place in 2019 and compared the simulated results to drone observations. In our simulations, we incorporated snow entrainment and by varying the snow characteristics in the path, mainly snow height and temperature, we demonstrate the complex relationships between initial snow conditions (volume and temperature), the various processes that take place during avalanche flow (frictional heating, entrainment, etc.), and the resulting flow behaviour, for which runout length and impact pressure are of highest concern for road safety managers. We also demonstrate how various flow regimes may develop, and how some interesting flow behaviours may be expressed, based on changes in the snowpack in the path.

The applied model continues to utilise a Voellmy-based frictional approach as in the well-established models applied for hazard mapping. However, the friction coefficients are now dynamically calculated and are affected by terrain and snowpack variables, which differ considerably between different avalanche tracks and avalanche periods. To set up the simulations we apply snowpack parameters measured at nearby automated weather stations and snow profiles. Further, we utilised photogrammetrically measured snow depth distributions acquired by drones, that were able to capture the extreme spatial variability of snow depth distribution in our avalanche paths of interest.

The results indicate that we can use measurements from weather stations at different altitudes and locations to calculate the snow cover distribution and snow temperature gradient and let those inform our model to simulate realistic avalanches. Choosing weather stations from a nearby valley (approximately 3 km distance) showed acceptable results with a coefficient of variation below 5 %. Comparing the modelled avalanche outlines to the observed ones, it is visible, how the model represents

important features such as the evolution of fingers indicating zones with higher impact pressures or the development of the powder cloud.

In this publication, the version RAMMS::EXTENDED 2.8.28 was analysed in more details. It is important to point out, that the model will be further developed and calibrated. Hence, the presented parameter set must be treated carefully and can only be applied to this version. For more current versions, the publication by Stoffel et al. (2024) will give more insights.

The presented approach will now be applied to calculate avalanche runout for different representative weather and snowpack scenarios for the Dischma road. These results will then be evaluated by local experts and the applicability of this approach for future decision making will be assessed. A probabilistic approach is currently being tested to calculate reach probabilities to the road for specific avalanche tracks. These are important steps towards a more data-based decision making for road management in mountain regions. In the future, this concept could also be applied to assess additional areas which are endangered by avalanches, such as houses build in hazard zones and ski slopes.

## 6 Outlook

Building on the foundations of this project, we aim to develop a tool that integrates real-time data with numerical simulations to predict the daily avalanche runout distance. To achieve this objective, the following directions can be pursued in future work.

As nearby and representative weather station measurements are not accessible for many roads, in a next step, snow cover models as SNOWPACK (Lehning et al., 1999) and CROCUS (Vionnet et al., 2012) could be used to calculate the relevant input parameters. For rough estimations of the snow temperature, it would be necessary to compare the cloud coverage during the hours before the avalanche occurred to the average temperature of the release snow mass. As the runout distance is not too sensitive to the snow temperature, a first approach could be to define simulation scenarios of cold and warm temperatures depending on the cloud coverage.

To enable the model to simulate additional wet snow avalanches, we need to collect more avalanche data that is directly related to measured snowpack temperature and moisture content.

In the currenty project we worked with a given release zone which we got from the drone data. In an application for practice, we would additionally need to make an expert guess on where the potential release areas could be. This is an important source of uncertainty. A first approach could be to use automatically delineated release areas as proposed by (Bühler et al., 2022, 2018).

To advance the development of a practical tool, additional factors beyond avalanche runout distance must be considered. First, the system should incorporate the probability of avalanche release under daily conditions, with an initial approach being to integrate the avalanche danger level into the calculations. Second, our analysis highlights the significant influence of snow entrainment on avalanche dynamics. Observations from past avalanche events Issler et al. (2019) indicate that accurately estimating the potential erodible snow is essential. In a simplified model, erodible snow height could be approximated by identifying weak layers in the SNOWPACK model.

Furthermore, a method for displaying relevant data to experts must be designed. Key questions will arise as how to establish thresholds for acceptable powder cloud pressure or snow depth on roads, as well as the acceptable probability levels for road openings.

*Acknowledgements.* This research is funded by the Swiss National Science Foundation (SNSF) under project number 207519, titled "Avalanche Safety for Roads". The authors express their gratitude to SOS Davos Klosters for their collaboration and provision of avalanche run-down pictures. Additionally, we want to express our gratitude to our reviewers for their thorough work and the thoughtful insights.

*Author contributions.* Expert numerical simulations measures: PB, JGA, KWJ Practitioner view: LS, RAMMS Implementation: MC, Manuscript: JGL with contributions from all co-authors

*Competing interests.* At least one of the (co-)authors is a member of the editorial board of Natural Hazards and Earth System Sciences.

*Financial support.* This research is funded by the Swiss National Science Foundation SNSF with the project "Avalanche Safety for Roads" (Nr. 207519).

*Data availability.* The post processing tool and the used data for the mapped avalanche outlines will be available on GitHub and EnviDat on the publication of the final script.

## Appendix

### 6.1 Snowpit Data

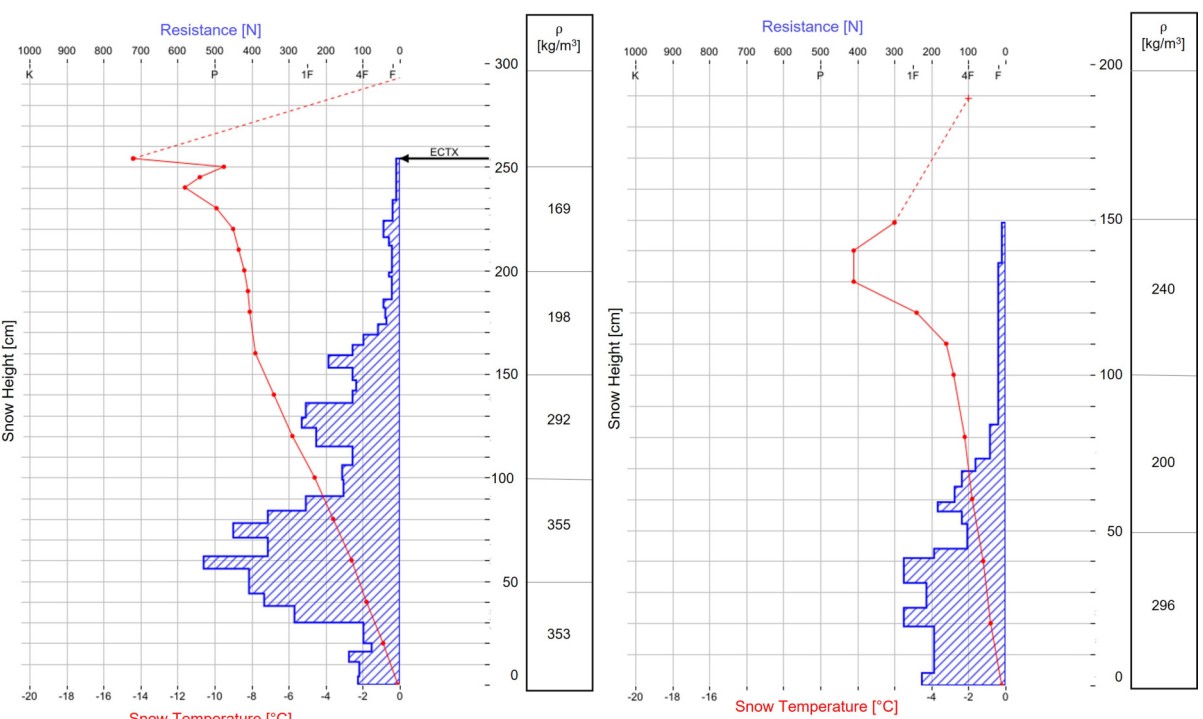

**Figure A1.** Snowpit data for the stations at Weissfluhjoch (left) and at SLF Davos (right) (Source: https://whiterisk.ch produced by SLF).

### Appendix.2 Closure Relations

In this subchapter we show the closure relations which result from calibrations based on data from Vallé de la Sionne. We want to point out that those relations represent the best fit with the data we have measured so far.

**Decay constant**

$$\beta = 1.40 + \frac{1.7}{\pi} \cdot 1.7 \cdot \mathrm{atan}\left[1.6\left(T_\Phi - 271.5\right)\right] \tag{25}$$

**Entrainment**

The mass eroded from the snowcover is described as follows:

$$\dot{H}_\Sigma = k_0 k_T k_\psi \|U_\Phi\|$$

where the parameters are chosen as follows

$$k_0 = 0.005 \qquad k_T = 3.00 - 0.6366 \cdot \mathrm{atan}\left[0.8 T_\Sigma - 213.6\right] \qquad k_\psi = \left[\sin(\psi) - \tan(\theta_b) \cdot \cos(\psi)\right] \tag{26}$$

$$\epsilon = f(T_\Sigma)g(r) = \left[ e^{\frac{-(T_\Sigma - 266)^2}{10}} \right] \left[ \frac{r^2 e^{\frac{-r^2}{2a^2}}}{a^3} \right] \tag{27}$$

with $r = \frac{h_\Phi}{h_\Sigma}$

and a = 1.1

**Air Into Cloud**

$$\dot{H}_{\Lambda \to \Pi} = \left( 1.16\psi + 0.013\sqrt{R_\Pi \hat{h}_\Pi} \right) (\rho_\Pi - \rho_\Lambda) \tag{28}$$

**Powder Cloud Drag**

The equation for the powder cloud drag $S_\Pi$ is given by:

$$S_\Pi = 0.1 \left[ \mu_L u_\Pi + 0.08\mu_T R_\Pi \hat{h}_\Pi \right] \tag{29}$$

where $\mu_L$ is the laminar drag and $\mu_T$ is the turbulent drag. Those coefficients are defined as functions dependent on a dimensionless drag coefficient $a_{drag}$ between 1 and 4 which is set by the user. The higher the dimensionless drag, the more turbulences there are in the powder cloud.

$$\mu_L = -0.22 + 0.28a_{drag} \tag{30}$$

$$\mu_= 0.02 + 0.035a_{drag} \tag{31}$$

The range of the powder cloud drag depends on the mean speed and the total turbulent energy. For a lower decay number $\beta_\Pi$, the cloud drag is higher which results in a smaller cloud flow width. In case of a higher number $\beta_\Pi$, the powder cloud drag will be lower and the cloud flow width will be larger.

**Dispersive Pressure**

The concept of dispersive pressure is described in more detail in the publication Buser and Bartelt (2011). It arises from the shearing in the core, which induces random motion of the grains. Due to dispersive pressure, the core height $h$ increases, and the position of the centre of mass $k_z$ shifts in the z direction. The centre of mass is defined as being at half the core height:

$$k_z = \frac{h}{2} \tag{32}$$

Therefore, we describe the relation of the position, velocity and acceleration of the centre of mass by the following differential equations:

$$\mathcal{D}(t, h_z, \dot{h}_z, \ddot{h}_z) = \dot{h}_z(h_z, \dot{h}_z, \ddot{h}_z) \tag{33}$$

$$\left(\frac{\dot{h_z}}{2}\right)_t + \mathrm{div}(\frac{\dot{h_z}}{2}\boldsymbol{u}_\Phi) = \frac{\ddot{h_z}}{2} \tag{34}$$

$$\left(\frac{\ddot{h_z}}{2}\right)_t + \mathrm{div}(\frac{\ddot{h_z}}{2}\boldsymbol{u}_\Phi) = \frac{\dot{W_z}}{H_\Phi} - \left[g_z + \frac{\ddot{h_z}}{2}\right]\frac{\dot{h_z}/2}{h_z/2} \tag{35}$$

The average velocity of the core is denoted as $\boldsymbol{u}_\Phi$, $\dot{W}z$ represents the shearing work rate in the z-direction, and $M\Phi$ is the mass of the core volume. The differential equations are solved using a HILL finite volume scheme, as in the original version of RAMMS described in Christen et al. (2010).

In equation 35, we want to emphasize that we are working with a simplification, as the bed-normal component of gravity would directly determine the acceleration of the core's centre of mass (Issler et al., 2017).

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
