# Peer review of "Simulation of cold powder snow avalanches considering daily snowpack and weather situations"

_EGUsphere, 2024_

## Referee Comment (RC2)

Comments on "Simulation of cold powder avalanches considering daily snowpack and weather situations to enhance road safety"

by

Julia Glaus, Katreen Wikstrom Jones, Perry Bartelt, Marc Christen, Lukas Stoffel, Johan Gaume, and Yves Bühler" (2024):

General comments:
* * *
The authors present a study that investigates the predictive capabilities of an existing coupled dense & powder flow model within the RAMMS:Extended software. To achieve this, they include a formulation of snow temperature and a formulation of snow cover entrainment with a variable distribution of erodible snow into their model.

First at all, it is not clear if these formulations are new in this publication or arise from earlier publications about the extended version of RAMMS. For example, entrainment is included since at least Bartelt, 2012 or Bartelt, 2018a. And for the temperature formulation, I know at least of the work by Valero, 2015 which includs release temperature already. It would add to the paper's quality if some information about how these connect to this publication is added.

I did not check the model equations, partially because the multitude of symbols is rather confusing. What I do not find is how the snow temperature alters the friction law. Eq. 15 states, the only temperature dependent parameter is beta. It is also completely unclear why alpha has no temperature dependence; to my understanding, the generation of the powder cloud is clearly temperature dependent. A discussion of why such a relation is chosen is missing.

Most of the results and discussion of the paper are on a parameter variation for the model input (release temperature, entrainment and gradients thereof) on a synthetic topography. While this is valuable standard approach to check the model and effects of the implementation, it is not sufficient as a proof for the formulated model assumptions, i.e. many explanations about the simulation study in sec. 4.1 sound too much like one would know these processes and effects from real avalanches (line 320ff., line 339ff).

It is unclear to me if the chosen synthetic topography is suited. The main parameter is runout distance or avalanche length (which needs precise definition by the way), but you never give those distances on the topography. Where is 900m, where is 1600m? See comment below to Fig. 3. To my knowledge, the Bramabuhl avalanche track is very steep from top to bottom, and then completely flat. So, everything runs basically to the valley floor. How would the results change if you had chosen a parabolic profile?

The comparison and discussion of the simulation on real topography with the measured data is weak. I do not understand how these three avalanches can support the claimed temperature and erosion gradient model formulation. These three avalanches experience the same snow cover. And Sec. 4.2 actually fits the friction parameters to one of the avalanches. However, a good point is that all three avalanches can be modelled with the same set of friction parameters.

I generally do not understand why you do not use data from your own test-site Vallee de la Sionne. Mass balances and snow temperatures have been measured there for more than a

decade. Generally, the paper lacks comparison and discussion with recent advances and publications from experimental avalanche measurements.

Specific comments:
* * *
L53: reference to r.avaflow missing. Or do you mean the AvaFrame (2023) reference, which points to a different model?

Fig3: Right side: Please add color scale, potentially remove the black background, and add lines that indicate runout distance (or avalanche length?)

Fig4: Why did you swap the colormap direction between both panels? Please be consistent, see also Fig. 16. Add elevation contour lines, add map corner coordinates, draw avalanche outlines in left panel. The left panel is too small to see anything, what about having the left panel in textwidth and three smaller panels underneath showing each release area (and not only Ruechi). Where can I see the wind redistribution effects?

L109: What is this post-processing tool and where are max. velocity, flow height and pressure used? In the ISSW publication and in here only runout distance is presented. BTW: I like Fig.4 from the ISSW publication, would have helped in here as well (Reference entry is missing ISSW, year and URL)

Sec. 2.2: A precise definition of runout distance or avalanche length is missing. Please add.

Fig5: Right panel: The smooth temperature gradient shows how bad the rainbow colormap is: The temperature gradient appears to have sharp changes at -8 and -6 degrees. Not to mention red/green color blindness. (A suggested good read on this topic is "Choosing Colormaps in Matplotlib" or "Somewhere over the rainbow..." https://journals.ametsoc.org/view/journals/bams/96/2/bams-d-13-00155.1.xml)

Fig6: Why not include the values from the particular avalanche into the example box (from Tab1)?

L153: Are all three literatures relevant here? You are not dealing with water content as you state in the very next sentence.

L156: All your measurements are from flat fields, but you project the temperatures onto a north facing slope. Some reasoning of why this is applicable would be helpful

L160: Missing Z of runout area.

L165: Christen, 2010, is a reference to normal RAMMS. Choose one for the extended version.

L171: If all those citations are important, please elaborate what each contribution is. Please give a short statement on each or use them throughout the description of the model in this section.

L234: Tab 3.2 is not typesetted as a table, caption missing.

L245: The experimental results from the cited publication must be discussed in respect to your results later in the publication. I guess there are more suited literature that backs the sentence here (e.g. Ancey 2007 or others)

L297: Sec.4.1 should be snow temperature+entrainment and Sec.4.3 comparison to measured avalanches

Fig8: You may indicate location of Fig.9 at -6°C with vertical line. Why is the chosen value for Delta_T 0.3 instead of 0.5 as indicated in Tab.1?

L305: I don't see that Dent (1998) backs your statement here.

L320: This is written as a fact, but fundamentally a model assumption. I suggest to discussion with your colleagues from Vallee de la Sionne about powder snow avalanches.

L322: Why should -10°C be more realistic?

L323-326: Also quite speculative.

L334 and Fig9: wording d* maximum erosion depth or snow height, be consistent.

Fig10: Please use same temperature range for colormap in all 3 panels (0 -- -13°C), currently all three panels look basically the same. You may indicate the location of Fig.8 at 0.3°C/100m with a vertical line. Legend overlaps value of runout especially in middle panel.

L339: Which experience? I would say PSAs need a lot of new (erodible) snow but are rather independent of absolute snow height.

L342: Large avalanches can erode much more than 1 meter, e.g. Fig 16 the middle part of the track is snow free.

Fig11: Evolution over time can be shown with a plot over time. Your ISSW paper suggests you can easily extract runout distance for different time steps. Please show the results in a runout distance over time representation.

L348ff: Please extend the discussion incorporating the prior mentioned publication Steinkogler,2015, Fischer,2018, Kohler,2018, Li,2020, Ligneau,2022 to suggest some.

L352: Why show Wildi avalanche here and not Ruchi? What I understood is, that you use Ruchi as the main avalanche for Sec.4.2, so it would be good to show this avalanche in greater detail.

L358: Reaction to entrained snow temperature can be rather fast, see for example the rapid changes in front velocity reported by Kohler,2018.

Sec.4.2: By the way, what kind of parameter mu0, xi0, N0 are used in Sec.4.1?

Eq24: As stated in Tab3.2, xi(t) limits the flow velocity and mu(t) the runout. How to find a good xi, when no velocity data is used?

L382: You never elaborate on the true runout distance of the avalanches.

L384: Sentence is unclear. Explain in more detail.

Fig12: I believe the avalanche runout is at 1600-1650m, because this shows the best fit square. However, Fig 8, 9, 10 only show very few parameter constellations that are able to reach this far. Therefore, it seems the "high point" at -8°C in Fig8 is absolutely needed... Please comment/discuss this. Also, your best fit parameters are on the border of your parameter space study (lowest row), how does this matrix extend to values of mu0 > 0.55?

L393: Missing mentioning of Fig14 in text. Before it was Fig13 and now Fig15? Please extend discussion on Fig14, also include the drone data into Fig14 for comparison.

L393ff: Fig15 shows the height of the dense core and a picture of the cloud. How can we compare both and get a "good agreement"?

L396: How do you know the air blast never exceeded 5kPa?

F16: Please annotate the mentioned avalanche arm (L399). Do both panels show snow depth? Or does the simulation show the deposition depth? What is the bow shaped deposition in the simulation and why is the other deposition in the river/road only, and not as smoothly distributed as suggested by Glaus, 2023, Fig3? Please extend description and discussion on Fig16.

L405ff: Why is the simulated deposition volume outside the mapped area not taken into account?

L407ff: Sentence is unclear. Reformulate and extend.

L409ff: The sentence comes rather unexpectedly. Where do I see wind scour? What does that mean for the simulations? Have you run the simulation with less snow?

L415: Impact pressure of powder cloud does not play a role in this publication. I also doubt that it's the most relevant danger for cars and people.

L418: I don't believe the model assumptions are correct for temperatures near melting, and this already starts at -1 to -2°C. The Ruchi avalanche shows -8 to -5 degrees.

L429: Evolution of finger was not well discussed.

L433: Snow cover models are not needed to calculate cloud coverage. Snow cover models would directly give a snow temperature and more (see work of Wever,2016,2018). I don't believe that cloud coverage will alter the snow temperature more than 20cm into the snow cover, but relevant avalanches for the road have easily >1m release height.

As said earlier, please think of using the data of Vallee de la Sionne as a future outlook.

L448: JG, MC missing. Who is AV, AW, PRJ?

Carefully check reference again. My list may be incomplete.

L461: DOI missing

L464: DOI missing

L478: weird characters in german title. Is there any other publicly availiterature?

L480ff: entry twice

L484: DOI missing

L504: list of authors incomplete

L506: missing journal and year

L510: DOI missing

L513: DOI missing

L515: DOI missing

L523: reference to published version, not discussion

L530: DOI missing

L536: any publicly available resource?

L543: Journal and DOI missing

L548: reference to published version, not discussion

L545ff: consistently use either Vera Valero or Valero (I believe later one). Check duplicates (Valero,2015,2018 and Vera,2015,2018)

L557: reference to published version, not discussion

---

## Author Response (AR1)

**Response to Referees**
**Simulation of cold powder avalanches considering daily snowpack and weather situations to enhance road safety**

Julia Glaus, Katreen Wikstrom Jones, Perry Bartelt, Marc Christen, Lukas Stoffel, Johan Gaume, and Yves Bühler

November 30, 2024

Dear Editor and Referees,

We greatly appreciate your detailed and insightful feedback. This has been invaluable in enhancing the quality of the manuscript. We have copied your comments into the blue boxes and enumerated them so that we can address each part separately below. The responses to the comments from the first reviewer are marked as R1, and the responses to the comments from the second reviewer are marked as R2. Additionally, we have included our comments on the changes we have made in italics, providing the exact location in the manuscript.

Sincerely,

Julia Glaus & co-authors

**Response to Reviewer R2**

**General Comments**

> R2-1: The authors present a study that investigates the predictive capabilities of an existing coupled dense & powder flow model within the RAMMS:Extended software. To achieve this, they include a formulation of snow temperature and a formulation of snow cover entrainment with a variable distribution of erodible snow into their model.

> R2-2: First of all, it is not clear if these formulations are new in this publication or arise from earlier publications about the extended version of RAMMS. For example, entrainment is included since at least Bartelt, 2012 or Bartelt, 2018a. And for the temperature formulation, I know at least of the work by Valero, 2015 which includes release temperature already. It would add to the paper's quality if some information about how these connect to this publication is added.

We will include a more detailed description of the historical development of RAMMS Extended and highlight the differences and new features more clearly. The goal of the RAMMS chapter is to summarise the current state again and emphasise it with additional details compared to the existing literature for better understanding.

***L66:*** *In the first part of this paper, we provide an overview of the current version of the model, summarize recent literature on RAMMS::EXTENDED, and present the current calibrations.*

> R2-3: I did not check the model equations, partially because the multitude of symbols is rather confusing. What I do not find is how the snow temperature alters the friction law. Eq. 15 states that the only temperature dependent parameter is beta. It is also completely unclear why alpha has no temperature dependence; to my understanding, the generation of the powder cloud is clearly temperature dependent. A discussion of why such a relation is chosen is missing.

- We will adjust the symbols according to Dieter Issler's comments in point R1-8.2 to enhance the readability of the equations.

- The snow temperature is correlated with the internal energy and random fluctuation energy, which are influenced by the friction parameters.

- Your observation that alpha is temperature-dependent is certainly valid. We are working on a new version where alpha varies with temperature, but given the scarcity of calibration data around the critical transition between wet and cold avalanches, this is a challenging task. We aim to address this in the discussion, emphasising the simplifications made.

*L228:Due to limited data between cold and warm avalanches, a constant alpha of 0.07 is assumed for cold avalanches and 0.05 for warm avalanches. The calibrated curve for $\beta$ is shown in Appendix.2.*

> R2-4: Most of the results and discussion of the paper are on a parameter variation for the model input (release temperature, entrainment and gradients thereof) on a synthetic topography. While this is a valuable standard approach to check the model and effects of the implementation, it is not sufficient as a proof for the formulated model assumptions, i.e., many explanations about the simulation study in sec. 4.1 sound too much like one would know these processes and effects from real avalanches (line 320ff., line 339ff).

Our simulation is indeed based on mechanical assumptions and was calibrated using observed avalanches from the Vallée de la Sionne, along with adjustments made from data on avalanches worldwide. We acknowledge that a definitive proof for such a complex system is challenging to achieve, given the inherent uncertainties and variations in real avalanche events. But all depth averaged models make the assumption that the flow is attached to the topography with no slope perpendicular velocity. So there will be never proof for the system, but we can gain confidence by comparing our results to well documented real avalanches.

**First Paragraph on Avalanche Model**

> R2-5: It is unclear to me if the chosen synthetic topography is suited. The main parameter is runout distance or avalanche length (which needs precise definition, by the way), but you never give those distances on the topography. Where is 900m, where is 1600m? See comment below to Fig. 3. To my knowledge, the Bramabuhl avalanche track is very steep from top to bottom, and then completely flat. So, everything runs basically to the valley floor. How would the results change if you had chosen a parabolic profile?

1. We will add the definition of avalanche length.

2. Based on observations of avalanches at Brämabühl over recent years as shown in the graphic below, we would not agree that most avalanches reach the road.

3. We will conduct simulations using the parabolic profile. If these simulations result in significant differences, we will consider including them in the publication or at least in the appendix as an additional perspective.

1. *L119: We extract the outlines of both the core and cloud, determining the longest distance by identifying the two most distant points using the Convex Hull algorithm combined with the Rotating Calipers method (in Python via scipy.spatial.ConvexHull).*

2. *We decided to stay with the idealized plane in the publication as the idea was to keep roughly the steepness of the avalanche paths and just to show the sensitivity analysis without terrain effects. Out of interest we did first simulations on the parabolic profile as proposed in Avaframe for the analysis of the release zone length as shown in Figure R2.*

> R2-6: The comparison and discussion of the simulation on real topography with the measured data is weak. I do not understand how these three avalanches can support the claimed temperature and erosion gradient model formulation. These three avalanches experience the same snow cover. And Sec. 4.2 actually fits the friction parameters to one of the avalanches. However, a good point is that all three avalanches can be modelled with the same set of friction parameters.

The calibration of the model was done based on the VdlS data and not with the Brämabühl avalanches. This study focus more on the sensitivity. In the future, we will compare for sure with more avalanches beside VdlS as soon as we have good data.

*R2-2: we pointed this out in the manuscript as stated in R2-2.*

[Figure]

Figure R1: Observed avalanches at Brämabühl from 1964 to 2023. The left side of the graphic shows how many avalanches reached a certain distance. The right side shows the outlines of all observed avalanches. It can be seen that not all avalanches reach the valley, and most stop before the flat runout where the lower boundary of the forest is.

[Figure]

Figure R2: Representation of avalanche runout distance on a parabolic profile and on our idealized slope for Rüchi path.

> R2-7: I generally do not understand why you do not use data from your own test-site Vallée de la Sionne. Mass balances and snow temperatures have been measured there for more than a decade. Generally, the paper lacks comparison and discussion with recent advances and publications from experimental avalanche measurements.

We excluded the comparison with Vallee de la Sionne due to the counter slope. Additionally, the temperature data available for Valle de la Sionne are model-based (from SNOWPACK) rather than direct measurements. We decided for our test region, as all three avalanches have an unconfined run-out. To motivate our test region in the publication, we can add this "train of thoughts".

**Specific Comments**

> R2-8: L53: Reference to r.avaflow missing. Or do you mean the AvaFrame (2023) reference, which points to a different model?

We will add both models incl the reference. **L58:** *we added reference to Avaframe model.*

> R2-9: Fig3: Right side: Please add color scale, potentially remove the black background, and add lines that indicate runout distance (or avalanche length?)

We will implement this as proposed. ***Figure 4:*** *we added a new figure showing the slope angle and a side view of the idealized plane.*

> R2-10: Fig4: Why did you swap the colormap direction between both panels? Please be consistent, see also Fig. 16. Add elevation contour lines, add map corner coordinates, draw avalanche outlines in left panel. The left panel is too small to see anything, what about having the left panel in textwidth and three smaller panels underneath showing each release area (and not only Ruechi). Where can I see the wind redistribution effects?

- We will unify the colour scaling according to Dieter Isslers inputs R1-8.1

- Adding counter lines we can try - but I expect it to reduce the readability of the figure.

- We can add the figure for all three avalanches

***Figure3*** *we adjusted the color scale and added a graphic of all three release zones. Adding couture line made the picture "oversaturated" and it was hard to read the graphic.*

> R2-11: L109: What is this post-processing tool and where are max. velocity, flow height and pressure used? In the ISSW publication and in here only runout distance is presented. BTW: I like Fig.4 from the ISSW publication, would have helped in here as well (Reference entry is missing ISSW, year and URL)

- The post processing tool is an in-house written python code that we can publish on Github

- Max velocity, flow height and pressure are used for comparison to the observed avalanches

- We try not to overload the publication - but we can add a reference to the (Glaus et al., 2023)

***Code and Data:*** *uploading code to Githup publishing avalanche outlines, release zones and additional data as velocity and hight in progress.*

> R2-12: Sec. 2.2: A precise definition of runout distance or avalanche length is missing. Please add.

As answered in R2-4.

> R2-13: Fig5: Right panel: The smooth temperature gradient shows how bad the rainbow colormap is: The temperature gradient appears to have sharp changes at -8 and -6 degrees. Not to mention red/green color blindness. (A suggested good read on this topic is "Choosing Colormaps in Matplotlib" or "Somewhere over the rainbow..." https://journals.ametsoc.org/view/journals/bams/96/2/bams-d-13-00155.1.xml)

We can recolour the graphic to improve readability for colour-blind people. However, please note that the current version is the standard output from RAMMS, where I cannot adjust the basic colour scaling. **Overall:** *We adjusted the color scalings where it was possible.*

> R2-14: Fig6: Why not include the values from the particular avalanche into the example box (from Tab1)?

We will implement this as proposed. **Figure 6** *we adjusted the Figure with the values from the Rüchi case.*

> R2-15: L153: Are all three literatures relevant here? You are not dealing with water content as you state in the very next sentence.

Yes

> R2-16: L156: All your measurements are from flat fields, but you project the temperatures onto a north facing slope. Some reasoning of why this is applicable would be helpful.

This is the traditional way that even the avalanche forecasting team is doing it. We are very interested in input on how to do it differently. We put a measurement station in an avalanche release zone, but it was destroyed by an avalanche. We know that the information is not perfect from the flat fields - but it is a safe approach. We performed additional measurements with thermal cameras confirming the validity of this assumption.

> R2-17: L160: Missing Z of runout area.

We will implement this as proposed ($Z_{runout} = 1600 m.a.s.l$). **L179:** *we added the information to the description.*

> R2-18: L165: Christen, 2010, is a reference to normal RAMMS. Choose one for the extended version.

In the first sentence we talk about the original RAMMS model - so we keep the Christen reference. We can add a sentence that the latest publication on the new version was done by Zhuang et al 2023 and add this reference. **L185:** *The basics of the model are presented in (Zhuang et al., 2023a).*

> R2-19: L171: If all those citations are important, please elaborate what each contribution is. Please give a short statement on each or use them throughout the description of the model in this section.

This point will be covered by working on your comment R2-2.

> R2-20: L234: Tab 3.2 is not typesetted as a table, caption missing.

Will be adjusted.
**Table2:** *Adjusted.*

> R2-21: L245: The experimental results from the cited publication must be discussed in respect to your results later in the publication. I guess there are more suited literature that backs the sentence here (e.g. Ancey 2007 or others)

We can do this.

> R2-22: L297: Sec.4.1 should be snow temperature+entrainment and Sec.4.3 comparison to measured avalanches

We will implement this as proposed. **L350** *We adjusted the sentence along the new section structure.*

> R2-23: Fig8: You may indicate location of Fig.9 at $-6°C$ with vertical line. Why is the chosen value for $\Delta T$ 0.3 instead of 0.5 as indicated in Tab.1?

- We can add the vertical line
- We can add the evaluation of $\Delta T$ equals 0.5 to have a more unified approach. The 0.3 was chosen as it is a commonly used value in the Alps. Overall, it will not influence the discussion.

[Figure]

**Figure 8** *We show now always the parameter set for Rüchi path.*

R2-24: L305: I don't see that Dent (1998) backs your statement here.

Correct, we mistaken with the reference. Alternatively, we can refer to the paper (Li et al., 2020)
**L360:** *.. while warmer avalanches will exhibit more plug-type flows (Li et al., 2021)*

R2-25: L320: This is written as a fact, but fundamentally a model assumption. I suggest to discussion with your colleagues from Vallee de la Sionne about powder snow avalanches.

We will reformulate to "We assume .."
**L375:** *We assume the reason for this behaviour is the decay parameter of granular temperature $\beta$.*

R2-26: L322: Why should -10°C be more realistic?

In the alps we don't observe normally snow covers which show an average snow temperature far below $-10°C$. Therefore, checking the temperature above $-10°C$ brings us in a region of the diagram where we represent more realistic the observed conditions.

R2-27: L323-326: Also quite speculative.

We describe our observations and try to give an interpretation of our investigation.

R2-28: L334 and Fig9: Wording d* maximum erosion depth or snow height, be consistent.

We will go with maximum erosion depth.
**Manuscript:** *we adjusted the wording in the manuscript.*

R2-29: Fig10: Please use same temperature range for colormap in all 3 panels (0 – -13°C), currently all three panels look basically the same. You may indicate the location of Fig.8 at 0.3°C/100m with a vertical line. Legend overlaps value of runout especially in middle panel.

- We will unify the colour scaling according to Dieter Isslers inputs R1-8.1

- We will mark the 0.3°C/100m

R2-30: L339: Which experience? I would say PSAs need a lot of new (erodible) snow but are rather independent of absolute snow height.

Yes, the way the model is run, we use the new snow to drive the erodible layer according to the Swiss Guidlines. In the ISSW abstract 2024 we discuss more details on

[Figure]

Figure R3: Runout distance over time of Rüchi path for the parameter set given in the main publication.

R2-31: L342: Large avalanches can erode much more than 1 meter, e.g., Fig 16 the middle part of the track is snow free.

Exactly, we will adjust the phasing to point out that this is a case specific statement.
**L397:** *we adjusted the statement, such that it is clear that this results only holds true for this specific set of input parameters.*

R2-32: Fig11: Evolution over time can be shown with a plot over time. Your ISSW paper suggests you can easily extract runout distance for different time steps. Please show the results in a runout distance over time representation.

Sure, we can add this.
**R3:** *We added the figure here in the review answers.*
  **Evaluation:** *Below we show you the figure where we measured the avalanche length for ever 5s.*

R2-33: L348ff: Please extend the discussion incorporating the prior mentioned publication Steinkogler,2015, Fischer,2018, Kohler,2018, Li,2020, Ligneau,2022 to suggest some.

Sure, we can add this.

R2-34: L352: Why show Wildi avalanche here and not Ruchi? What I understood is, that you use Ruchi as the main avalanche for Sec.4.2, so it would be good to show this avalanche in greater detail.

We can replace the graphic with Ruechi Tobel. Most of the investigations are primarily focused on Wildi, as we are also conducting the measurement campaign there (Ruttner-Jansen et al., 2023). In comparison, Ruechi Tobel is a simpler avalanche to evaluate because it doesn't split into two arms, making it easier for a general assessment.
**Manuscript:** *We unified now the analysis and focused on the Rüchi path.*

R2-35: L358: Reaction to entrained snow temperature can be rather fast, see for example the rapid changes in front velocity reported by Kohler,2018.

We can add this fact and citation.

R2-36: Sec.4.2: By the way, what kind of parameter mu0, xi0, N0 are used in Sec.4.1?

We used a mu0 of 0.55; xi0 of 1800 and a N0 of 200 as described in the table on page 13 (caption will be added to the table).
**L354:** *As a base we used the model parameters shown in Table 2.*

R2-37: Eq24: As stated in Tab3.2, xi(t) limits the flow velocity and mu(t) the runout. How to find a good xi, when no velocity data is used?

We checked with the VdlS avalanches the friction values from the swiss guidlines. We tried out the same friction values for RAMMS extended and checked with the velocities from the radar measurement. As we still got good results, we fixed those values.

*Figure 2: We added the velocity data we have from the Brämabühl event. Those are high level assumptions, but they are in good agreement with the simulations as shown in Figure 14.*

R2-38: L382: You never elaborate on the true runout distance of the avalanches.

We can add this as a discussion point.
*Table 1: We added the runout distances in Table 1.*

R2-39: L384: Sentence is unclear. Explain in more detail.

We will rephrase this sentence. The initial calibration of RAMMS::EXTENDED was based on the Vallée de la Sion avalanches, where velocity data was also included. We aimed to reuse those friction values to determine if they yield realistic results for much smaller avalanches, such as those at Brämabühl.
*Manuscript: We added overall more details about the velocity data in the manuscript to clarify this point.*

R2-40: I believe the avalanche runout is at 1600-1650m, because this shows the best fit square. However, Fig 8, 9, 10 only show very few parameter constellations that are able to reach this far. Therefore, it seems the "high point" at $-8°C$ in Fig8 is absolutely needed... Please comment/discuss this. Also, your best fit parameters are on the border of your parameter space study (lowest row), how does this matrix extend to values of mu0 $> 0.55$?

Yes, this result arises because we used a different temperature gradient for this simulation. We will ensure that all simulations are conducted with the parameter set identified for the Brämabühl avalanches to enhance consistency and clarity. We can do the simulation again for mu0 $> 0.55$ - for this paper we defined the interval for mu0 by checking values which we normally used. Therefore the values from the back calculations are not in the centre of the diagram.

R2-41: L393: Missing mentioning of Fig14 in text. Before it was Fig13 and now Fig15? Please extend discussion on Fig14, also include the drone data into Fig14 for comparison. Fig15 shows the height of the dense core and a picture of the cloud. How can we compare both and get a "good agreement"?

We will include Fig14 in the text and add the drone data. We focused in the comparison on the spread of the cloud - we will work here again on the comparison.
*Figure 12 and L459: We included now the figure in the text.*

L396: How do you know the air blast never exceeded 5kPa?

The powder cloud reached the barns, but we did not observe any damage to the windows or other infrastructure that got damaged.
*L469: .. as houses and windows remained undamage.*

F16: Please annotate the mentioned avalanche arm (L399). Do both panels show snow depth? Or does the simulation show the deposition depth? What is the bow shaped deposition in the simulation and why is the other deposition in the river/road only, and not as smoothly distributed as suggested by Glaus, 2023, Fig3? Please extend description and discussion on Fig16.

Sure we can annotate the avalanche arm. The left pannel shows the measured snow depth and the right pannel the deposition height. In Glaus 2023 we used a different parameter set and showed the snow height of the last calculation step vs in this graphic the deposition height. We will work on the discussion.
*Manuscript: We clarified in all figures whether we show a deposition value or a time step.*

L405ff: Why is the simulated deposition volume outside the mapped area not taken into account?

In the area where the avalanche traverses the forest, our remote sensing measurement method reaches its limits. The presence of trees creates gaps in our snow measurements, leading to unreliable data in those regions. Therefore,

we confined our comparison to the run-out area, where we are confident that our measurements accurately reflect the conditions.

> L407ff: Sentence is unclear. Reformulate and extend.

We will reformulate.
*Anwered with review comment L405ff.*

> L409ff: The sentence comes rather unexpectedly. Where do I see wind scour? What does that mean for the simulations? Have you run the simulation with less snow?

Yes, we also conducted simulations using the snow measurements from individual avalanche tracks. We can provide these results to the reviewer if needed. We chose not to include them in the publication to avoid extending its length.

> L415: Impact pressure of powder cloud does not play a role in this publication. I also doubt that it's the most relevant danger for cars and people.

It strongly depends on the region. If you have trucks on the road, the impact pressure will not be negligible.

> L418: I don't believe the model assumptions are correct for temperatures near melting, and this already starts at -1 to -2°C. The Ruchi avalanche shows -8 to -5 degrees.

Correct, as the title of the publication states we focus on cold avalanches with this model. For warm avalanches, we don't have enough calibration data.

> L429: Evolution of finger was not well discussed.

We will improve the discussion.

> L433: Snow cover models are not needed to calculate cloud coverage. Snow cover models would directly give a snow temperature and more (see work of Wever,2016,2018). I don't believe that cloud coverage will alter the snow temperature more than 20cm into the snow cover, but relevant avalanches for the road have easily >1m release height.

Let's give it a try. This approach would be an unconventional method if we don't have a snowpack simulation or sensors in the snow cover. This also means that we would accept a larger margin of error. On a cloudy night, the temperature profile within the snowpack is quasi constant. Consequently, it might be possible to estimate the temperature using a surface sensor. As I mentioned before in the Alps we don't observe a very large range of average temperature values in the snowcover. Hence, we can at least try to get a rough approximation.

> Comments on references

We will clean up the biblography and include all your inputs.

**References**

Glaus, J., Wikstrom-Jones, K., Buehler, Y., Christen, M., Ruttner-Jansen, P., Gaume, J., and Bartelt, P.: RAMMS::EXTENDED – Sensitivity analysis of numerical fluicized powder avalanche simulation in three-dimensional terrain, doi: https://arc.lib.montana.edu/snow-science/objects/ISSW2023_P2.26.pdf, 2023.

Li, X., Sovilla, B., Jiang, C., and Gaume, J.: The mechanical origin of snow avalanche dynamics and flow regime transitions, The Cryosphere, 14, 3381–3398, doi: 10.5194/tc-14-3381-2020, 2020.

Ruttner-Jansen, P., Glaus, J., Wieser, A., and Buehler, Y.: A Measurement System for Mapping Snow Distribution Changes in an Avalanche Release Zone, 2023.

**Response to Referees**

**Simulation of cold powder avalanches considering daily snowpack and weather situations to enhance road safety**

Julia Glaus, Katreen Wikstrom Jones, Perry Bartelt, Marc Christen, Lukas Stoffel, Johan Gaume, and Yves Bühler

November 30, 2024

Dear Editor and Referees,

We greatly appreciate your detailed and insightful feedback. This has been invaluable in enhancing the quality of the manuscript. We have copied your comments into the blue boxes and enumerated them so that we can address each part separately below. The responses to the comments from the first reviewer are marked as R1, and the responses to the comments from the second reviewer are marked as R2. Additionally, we have included our comments on the changes we have made in italics, providing the exact location in the manuscript.

Sincerely,

Julia Glaus & co-authors

**Response to D. Issler**

**0.1 Major Remarks**

> **R1-1: Title:** The present title of the manuscript relates more the vision and long-term goal of the authors rather than the actual content of the work they report on. What they show is that, for a given day when there was high or very high avalanche danger and artificial avalanche release led to three events in the study area, their proposed procedure for estimating avalanche runout would have been able to predict the extent of the run-out areas of these specific avalanches quite well. This is a significant achievement, but to achieve the stated goal of using weather and snow pack data daily to enhance road safety, the method should be demonstrated to work in other situations as well (and preferably also in other places). This discrepancy can easily be removed by adjusting the title and a few text passages, however.

Thank you very much for your suggestion on broadening the scope of the paper. We removed the road safety aspect in the title, resulting in "Simulation of cold powder avalanches considering daily snowpack and weather situations". *Title: adjusted to "Simulation of cold powder avalanches considering daily snowpack and weather situations" and comment in introduction that topic can be applied more broaden.*

> **R1-2: Project goal:** The following remark concerns the very choice of the research topic, and the authors need not address it in the revised version of the manuscript, but it is perhaps worth a thought. The work leading to this paper was financed by a project focusing on improving road safety. However, if the objectives of the manuscript are achieved, the method would be of at least equal use when it comes to deciding whether to evacuate specific settlements because of imminent avalanche danger or not—a situation recurring yearly in Norway. Thus, the focus on road safety in the title indicates an unnecessarily constrained scope. Furthermore, I would venture the guess that the majority of substantial avalanche events in paths that threaten roads in the Alps will reach that road with high probability. The primary question for the road safety managers is therefore about the release probability, not about the run-out. This question is not addressed in this paper, however. Moreover, in Switzerland the local avalanche commissions typically know from experience how far avalanches are likely to reach in a given situation, and they will not risk keeping a road open if the simulation indicates that the avalanche will stop 50 m before the road. The uncertainty in the calculation of avalanche run-out is, in my opinion, still larger than we "experts" often make ourselves believe.

Many thanks for the suggestion. We will discuss this aspect further in the introduction.

*L539: To advance the development of a practical tool, additional factors beyond the avalanche runout distance must be considered. First, the system should incorporate the probability of avalanche release under the daily conditions. A first approach could involve integrating the avalanche danger level into the calculations. Furthermore, a method for displaying relevant data to experts must be designed. Key questions will arise as how to establish thresholds for acceptable powder cloud pressure or snow depth on roads, as well as the acceptable probability levels for road openings.*

> R1-3: **Fracture depth:** It is understandable why the authors try to adhere to the method for selecting the fracture depth outlined in the Swiss guidelines. However, this may run into problems in the present context: While it is true that the majority of dry-snow avalanche releases involve only the new-snow layer, there is a non-negligible fraction of events that release on a deeply buried weak layer. For example, two very big powder-snow avalanches with average fracture depths of 1.5–2 m occurred at the Albristhorn and at Scex Rouge in Switzerland in January and February, 1995 after a very moderate snowfall (Issler et al., 1996, 2020). If the decision on road closure had been taken on the basis of a simulation with a fracture depth corresponding to the new snow (10–20 cm), the road below the Albristhorn would have remained open, with potentially dire consequences. I do not mean to require that the authors revise their method to encompass this issue, but it is important that they point out this potential weakness,which needs to be addressed before the method and the model can be used safely for such decision

This definitely holds true. We discussed this point in the ISSW extended abstract 2024. We will highlight this point additionally in the discussion.

*L541: Second, our analysis highlights the significant influence of snow erosion on avalanche dynamics. Observations from past avalanche events Issler et al. (2019) indicate that accurately estimating the potential erodible snow is essential. In a simplified model, erodible snow height could be approximated by identifying weak layers in the SNOWPACK model.*

> R1-4: **Studied parameter dependencies:** In my opinion, extra insight could be gained if the authors also varied the fracture depth, the release-zone length and the slope angle in the simulations on the idealized profile. This could presumably be done with little additional effort. The authors' statement that changes in available new-snow depth beyond 1 m do not influence the run-out distance may not be valid under more extreme conditions. A robust procedure for (largely automated) simulations for site-specific forecasting and warning can only be achieved if all relevant dimensions of the parameter space have been explored and included in the recommendations.is presumably dependent on the release mass or fracture depth. In simpler models, non-dimensionalizing the equations helps in identifying the parameters that govern the behavior of the system, but in the present case it remains to be seen whether this leads to additional insight. Again, I do not insist that the authors do all of this in the present manuscript, but I would like to encourage them to put "more meat on the bone" to make the paper more conclusive and valuable. At the very least, a brief discussion of these matters should be added

Many thanks for this input - we will include this analysis for the idealized profile.

*Chapter 4.2: We added a full chapter on the analysis of the release zone to investigate the influence of the release area length and steepness on the avalanche runout on an idealized plane. In Fig. R1 we additionally show the evaluation given for the alpha angle.*

> R1-5.1: **RAMMS::EXTENDED:** This code plays a pivotal role in the proposed methodical approach, and some parts of the model are explained in detail. However, neither the all-important mass exchange terms between the suspension layer and the dense core and the ambient air nor the source term for the height evolution in Eq. (7) are specified. These source terms must be modeled in terms of the dynamical variables of the system, which introduces additional parameters, which influence the behavior of the avalanche, as is shown, e.g., in (Vicari and Issler, Ann. Glaciol., 2024) but the values of which are not given a priori. The authors effectively pin down these parameters to some unspecified values without studying the sensitivity.

The reviewer is correct that the formation of the cloud from the core is the result of a complex interaction between the core, the ambient air and the generation of turbulence. Our goal in this paper was to express general physical relationships between the core, the cloud, the snow layer and ambient air as our application is directed toward road safety. We will add more details to the mass exchange terms between the suspension layer and the dense core and

[Figure]

Figure R1: figure
Representation of the avalanche runout length if the release length and the release angle are varied for the core and the cloud. In grey, the release volume is represented.

the ambient air.

**Attachment** *We have added the equation for air that gets into the cloud in the attachment. The term is calibrated based on avalanches from Vallée de la Sionne.*

> R1-5.2: Since the source terms are not specified while RAMMS::EXTENDED apparently has been under incremental development, it is not clear to which degree the model still corresponds to the stages described in earlier papers by Bartelt and coworkers. Since the 2023 GRL paper by Zhuang et al. appears most up-to-date, it would be preferable to mainly refer to that paper, in which the other references are also cited. I presume the authors wanted to keep the manuscript focused on the new work rather than the description of the model. I feel, however, that the present balance is not optimal—many equations are shown, yet many closure relations are kept hidden and have to be looked up in other papers. The authors might want to consider to either reduce the model description to one or two paragraphs succinctly characterizing RAMMS::EXTENDED, or to add the missing relevant pieces of information to make the paper reasonably self-contained. (My preference would be the latter if most of the following criticisms are addressed.)

We agree that including the closure relations would enhance the clarity of the RAMMS::EXTENDED model. We will add statements in the main text indicating where closure relations are applied, and provide detailed explanations of these in the appendix. It is important to note that these relations are primarily derived from calibrations based on observed avalanches, representing those that yielded the best results with the data we have measured so far.

**Attachment** *We have added the closure relations in the attachment for the decay constant, erosion and drag.*

> R1-5.3: In (Issler et al., J. Glaciol., 2018), several central aspects of RAMMS::EXTENDED were analyzed and shown to have mathematical or physical inconsistencies. One of the issues raised then has apparently been corrected: The gravitational force on the suspension layer is now included. However, the authors' statement that it is generally negligible is not correct: It does indeed apply as long as the dense core and the suspension layer travel together, but in the runout zone of the latter—after the dense core has stopped—and particularly when climbing a counter-slope, this term is of paramount importance if the suspension layer is well-developed. Moreover, the hydrostatic pressure gradient, which drives much of the lateral spreading of the cloud, is due to the excess buoyancy, i.e., to gravity.

We will remove this statement and open a new section in the discussion where we highlight those points to show more which assumptions are taken with this model.

**L298:** *We added the statement: Generally, we observe $\dot{M}_{\Phi\to\Pi}\vec{u}_\Phi \gg \left(\frac{\rho_\Pi-\rho_\Lambda}{\rho_\Pi}\right)\mathbf{G}_\Pi$. This observation remains valid as long as the dense core and the suspension layer move together. However, once the core comes to a stop and the cloud ascends a counter slope, the gravitational term will once again become more significant, as discussed in (Issler et al., 2017).*

> R1-5.4: Another major criticism in (Issler et al., 2018) concerns the equation of motion of the dense core in the bed-normal direction, for which Bartelt and Buser arrived at a third-order PDE in time for k, the z-coordinate of the center-of-mass of (Langrangean) control volumes, with the time derivative $\dot{R}_K$ of the granular temperature, $R_K$, as the source term. Instead, $R_K/m - g_z$, with $g_z$ the bed-normal component of gravity, should directly determine $\ddot{k}$. I do not expect Perry Bartelt and myself ever to come to agreement on this point, but it would be in the interest of the readers to alert them to this controversy (unless this part of the code has tacitly been changed, in which case it would be important to state this as a modification of the published model).

We will add this discussion point to the same section as described in R1-5.3.

**L220:** *A critical discussion on the approach of modelling the acceleration of the bed-normal expansion of the core's centre of mass directly on the bed-normal component of gravity can be found in Issler et al. (2017).*

> R1-5.5: Equation (7), which describes the height evolution of the dense core, is written as a conservation equation. Since the explicit form of the source term $D(t)$ (actually, it should be $D(x,y,t)$) is not given in (Zhuang et al., 2023a) either, it cannot be determined whether the equation is correct. However, if $D$ comprises just the dispersive-stress terms, Eq. (7) must be written, not as a conservation equation, but as an evolution equation $(\partial_t + u_\phi * \nabla)h_\phi = D(...)$.

Bozhinski and Losev termed this equation a "volume conservation" equation. We explicitly calculate volumes of air that enter the core as the core expands and contracts. The equation tracks the center-of-mass of the granular ensemble. The equation we write is the equation we solve – for us, it is both a conservation (of air) and evolution (of the center-of-mass) equation. We can add this statement in the publication.

**Attachment:** *We added the system of equations in the attachment and added more details.*

> R1-5.6 In Lines 216–217, it is stated that the modified Voellmy model arises directly from chute experiments with flowing snow. This statement is misleading for readers who are not too familiar with granular mechanics because it appears to say that this is the correct form of the friction law. In (Issler et al., 2018), it is shown that this empirical model behaves very differently from true granular models based on kinetic theory or from the experimentally well-tested $\mu(I)$ model at very high speeds. In practice, it should be admissible to use this parameterization since there are extra parameters that must be fitted to observations. It is important, however, to be clear about the heuristic nature of the assumptions that are made.

We will rephrase the sentence from "This formula arises directly from chute experiments with flowing snow" into "This empirical formulation was calibrated based on chute experiments with flowing snow".

**Line 251** *This empirical formulation was calibrated based on chut experiments with flowing snow (Platzer et al., 2007b, a; Bartelt et al., 2015b).*

> R1-5.7 I do not understand Eq. (11). For one, the subscript i is not explained; from Eq. (10) in (Zhuang et al., 2023a), one can see that is meant. It is unclear where the time (increment?) $\Delta t$ comes from. It is said that "The term $\dot{Q}$ m represents the latent heat of melting ice", but $\dot{Q}$ m is a (latent) heat flux, so there is a mismatch of physical dimension. This passage must be clarified. From Eq. (10) in (Zhuang et al., 2023a), one can see that meltwater production requires $\dot{Q}_m > 0$ and thus $T_\Phi > T$. What seems to be missing is some sort of heat conductivity (which has dimension (time)–1 and allows to get rid of the ominous integral and $\Delta T$). This conductivity will, among others, depend on the mean particle surface-to-volume ratio.

Yes, thank you. The reviewer is correct. $T_i$ should be $T_\phi$. $T_i$ represent the temperature of the solid phase. We do not consider heat conductivity as we assume that the frictional heating process (the shearing/rubbing between the particles) and heat exchange with the air occurs at a much higher rate than the conduction of heat. The ominous integral simply states that all the specific heat energy above the melting is available to drive latent heat exchanges

in the time interval dt. This is a common approach in numerical codes, for example SNOWPACK which operates on the same principle to determine surface and sub-surface melting in snowcovers.

*L136 We adjusted the formula and explanation in the script.*

> R1-5.8 The mass exchange between the dense core and the suspension layer is assumed to be oneway. This is conceptually problematic because the vertical expansion of the core due to increased granular temperature cannot create a vacuum, hence some fraction of the air–snow mixture in the cloud must be sucked back into the dense core. This is discussed to some degree in (Vicari and Issler, 2024). I do not expect this effect to be major on the scale of an entire avalanche event, but the simplifying assumption should be clearly mentioned.

We will add this discussion point to the same section as described in R1-5.3.

*L300: Additionally, it is important to note that the mass flow between the cloud and core is assumed to be unidirectional. Consequently, the vertical expansion of the core, driven by increased granular temperature, prevents the formation of a vacuum within the core. As a result, a portion of the air-snow mixture from the cloud is inevitably drawn back into the core (Issler et al., 2024).*

> R1-5.9 An interesting and fairly advanced feature of the PSA model is the extra equation for turbulence. However, earlier one-layer 3D models like SL-3D, the two-layer 2D/3D model SAMOS-AT, the one-layer 1D model by Parker et al. (1986), the one-layer mass-point model by Gauer (1994) and the two-layer 1D model formulated in (Issler, 1998) used similar equations. It would therefore be appropriate to point out in which respect the authors' model differs from the older ones.

We will mention the models and add the reference.

*Line 304: The turbulence in the cloud due to the fluctuation energy are described by equation 19. The structure of the equation is based on earlier one-layer 3D models (Hermann et al., 1994; Gauer, 1995; Sampl and Zwinger, 2004).*

> R1-5.10 From Eq. (9) and Line 197, one sees that the random kinetic energy of the dense core receives a positive contribution from entrainment equal to a fraction $\epsilon_\phi$ of the kinetic energy imparted to the eroded mass per unit time, $\dot{L}_{\Sigma\to\Pi}$. In the suspension layer, it is tacitly assumed that the corresponding coefficient $\epsilon_\phi$ = 1. The chosen value of $\epsilon_\phi$ is not specified as far as I can see, and no explanation is given why $\epsilon_\Pi$ should be as large as 1.

In the cloud, air entrainment produces turbulence. As a textbook on turbulence we follow the work of Davidson (2015) which show the equivalence of entrainment and turbulence. Following Davidson we make the assumption that entrainment first produces turbulence, and then decays to heat. This was discussed in the work of Zhuang et al. (2023).

*L312 In the cloud, we assume inelastic collisions between particles, such that all energy is converted into random motion, with no energy dissipated as heat, as a simplification. Hence, the constant $\epsilon_\Pi$ is set at 1.*

> R1-5.11 The entrainment model Eqs. (20)–(23) assumes the entrainment rate to be proportional to the avalanche speed, without any reference to the stresses exerted by the avalanche on the snowcover. This is problematic because a very thin but fast layer (like the "splash" ahead of the avalanche front) would be able to entrain equally much as a 3 m thick flow. I implemented and tested this entrainment model in a code similar to RAMMS::AVALANCHE some ten years ago and found it to give completely unrealistic spreading of the avalanche under certain conditions. There are several papers showing that the entrainment rate should be proportional to the difference between the shear stress near the flow–bed interface, divided by the flow velocity (rather than proportional to it). With the Voellmy friction law, the drag term then indeed produces a contribution proportional to the speed, but only beyond the threshold given by the shear strength of the bed material. Again, I do not mind if the authors use this model presently as long as they alert the readers to its potential problems.

The reviewer is correct. We have a shear stress cutoff to limit the entrainment of small avalanches and will mention this in the text.

*322: In this model, even very thin avalanches could potentially erode the same amount of snow as a thick one (Issler et al., 2024). To prevent this, a shear stress cutoff is implemented, ensuring that an avalanche must exceed a certain energy threshold before it can erode snow.*

R1-5.12 Apparently, RAMMS::EXTENDED now assumes a linear density profile and a parabolic velocity profile (rather similar to SL-1D and MoT-PSA), but this is not mentioned at all. When comparing the simulated damage areas to the observed ones, it matters strongly whether these profile functions are applied or not, so this must be stated explicitly.

We will add the statement that we take the mean values for the density and velocity profile. The user must post-process the mean values.

**L204:** *The model assumes constant density and velocity profiles where the mean values are taken. Hence, to estimate damage areas, the profile function must be post-processed by the user.*

R1-5.13 Another issue left undiscussed is the assumption of constant splitting of highly dynamical quantities like $\dot{M}_\Sigma$ and several others. Also here, I do not mind if this simplification is made in the present stage of development, but it should be stated that this is not always a realistic assumption. For example, a slowly plowing wet-snow avalanche will have $\gamma \approx 0$ while this parameter might be much closer to 1 in a fast, strongly fluidized flow over fluffy new snow.

Yes, the reviewer is correct. When we simulate more moist avalanches we take $\gamma = 0$. This publication is focused on dry, powder avalanches. We will mention in the text that this is a simplification that we did.

**L346:** *For 3-day new snow accumulation periods we take for as a simplification $\gamma = 0.2$. For more moist avalanches, this value goes to zero.*

R1-5.14 An important point that is largely shoved under the rug in the manuscript concerns the large number of parameters in the model (many more than listed in Table 2). This is difficult to avoid with such a complex phenomenon, but it needs to be communicated clearly that many parameters, the values of which are poorly known, were kept fixed in this study and only three initial or boundary conditions were investigated. Testing the influence of the other parameters is a task for the (near) future and—in my opinion—a prerequisite for wide-spread use of the scheme in site-specific warning.

We went through a calibration phase with VdlS avalanches to evaluate those values. We can point out in the publication more the parameters which we calibrated. The procedures were presented older publications.

**L183:** *Given the complexity of avalanches, a large number of model parameters required calibration, but not all can be presented in detail in this publication due to space limitations.*

R1-6.1 **Referencing.** I have never before reviewed a manuscript with an equally high degree of selfcitation before—in more than half of the cited papers, at least one of the authors of this manuscript is a co-author. On the other hand, clearly relevant papers by other authors are left out. In the case of the early Russian work, the book by Bozhinskiy and Losev (1998) is cited, giving the wrong impression that those authors developed a powder-snow avalanche model in the late 1990s while it was Eglit who developed a 1D two-layer model with entrainment in the beginning of the 1980s. Her work was recently extended by Vicari and myself, and the corresponding paper (which has been known to the authors) contains a sensitivity study that addresses complementary aspects of the problem and is, in my opinion, quite relevant in this context.

We acknowledge that our current focus has been primarily on in-house literature, as this publication aims to summarize the foundational research behind RAMMS::EXTENDED. Moving forward, we will include a more comprehensive literature review. The work by Vicari and Issler was published two months after our paper was submitted, so while we regret not having access to it at that time, we will certainly incorporate its insights and citation in this publication.

**Overall paper:** *We corrected the fact on Eglits paper and added more literature research overall in the paper.*

R1-6.2 The authors took a significant fraction of the text in Sec. 3 more or less literally from earlier papers. Those papers are referenced, but the cut-and-paste passages are not marked as quoted text. I do not know how NHESS handles such cases, but several high-ranking Norwegian politicians recently had to leave their government posts and got their MSc titles revoked because of similar plagiarism in their theses. I would therefore recommend reformulating the corresponding passages or marking them explicitly as quotations.

We checked the manuscript using a commercial plagiarism tool and found no flagged passages beside some of the equations. As the paper builds on our ISSW 2023 work, some sections may resemble previous publications due to

the logical structure of explaining the equations, but we will revise these sections to avoid any concerns.

*L292 We expect the paragraph from line 292 was meant. We adjusted the text.*

R1-7.1 **Minor Remarks** The density of ice is 917 kg/m³, not 971 kg/m³. Interestingly, the same typo occurs in one of the cited papers, in a rather identical-looking sentence. Moreover, the authors state that the density of air is 1.225 kg/m³, but in reality this value varies considerably with temperature, pressure and humidity. Differences at the percent level will have little effect on the simulations, but it would be better to state clearly that a single typical value has been chosen. Please see the annotated manuscript for further small corrections and suggestions.

Thank you very much for pointing out, we can adjust this.

*L296 The density of air varies slightly with changes in temperature, pressure, and humidity, but this variation has a negligible effect on the simulation.*

R1-8.1 **Presentation** The manuscript is organised in a clear way and written in good English (mostly). There are, however, many passages that do not seem to have undergone careful review by the co-authors. In most cases, the work done is explained clearly, without being verbose. The main exception to this, as mentioned above, is that the all-important mass exchange terms between the dense core and the suspension layer and between the ambient air and the suspension layer are not specified. This should be added in the revised version.

The figures generally illustrate the points made in the text well, but the following improvements are desirable:

1. I did not understand the message of Figure 1 at all. Does it really contribute something essential to the paper?

2. In Fig. 2.a, it would help readers if the avalanche deposits were numbered. Also consider replacing Fig. 2.b, which is repeated as Fig. 15.b, by Fig. 3.a rotated by 90° counter-clockwise, so that it aligns with Fig. 2.a. Figure 3.b can stand on its own because it is not visibly related to Fig. 3.a.

3. In Fig. 6, $d_0^*$ is drawn in the vertical direction rather than normal to the ground. If this is correct, this should be mentioned in the text, otherwise it needs to be corrected in the figure.

4. The exchange term $\dot{M}_{\Phi \to \Pi}$ in Fig. 7 is characterized as "air blow-out". If this were "just air" (to cite the late Othmar Buser), where would the snow in the suspension layer come from?

5. Figures 8 and 9 would be easier to read in one-column layout, i.e., reduced in width. Also consider lines connecting the points in Fig. 8 as in Fig. 9. Note that the keys are hard to read.

6. The message of Fig. 10 would be easier to understand if all three plots had the same color scale going from $-12\,°C$ to $0\,°C$. Make sure the figure keys do not hide the data points (a single figure key outside the plots themselves would suffice).

7. Color nuances for values close to 1600 m are hard to distinguish in Fig. 12. A different palette could help. Also, neither the text nor the figure state which value of the run-out distance was actually observed.

8. In the legend of Fig. 14, it ought to be mentioned that this is the result of the simulation.

9. Readers would be particularly interested in a map of the suspension-layer pressure for the simulation in Fig. 15 (perhaps add corresponding maps for the other two paths as well).

10. In Fig. 16, it is unclear for readers whether this deposit comprises only the dense core or includes the suspension-layer deposits as well.

Point 1: as we will take away the focus of this publication from road safety, we can remove this graphic. Point 3: $d_0^*$ is perpendicular to the surface. we will point this out in the text and adjust the graphic Point 10: We will mention that this only includes the deposition of the dense core. Points 2,4,5,6,7,8: We will implement these points to make the graphs more uniform and readable.

*1. Removed*

2. *Deposits numbered in Figure 1. 2b now only in Figure 14.*

3. *We marked the 90∘ angle in the graphic*

4. *We adjusted the graphic to show that it is a mixture of air and snow dust that is blown-out*

5. *Adjusted in Figure 8*

6. *Adjusted in Figure 9*

7. *We adjusted the color scale (now Figure 12). The runout distances for all three avalanches are added in the Table 1 and additionally marked in the Figure 13.*

8. *Added in caption*

9. *We adjusted the graphic showing now the powder cloud height, pressure, velocity and cloud density.*

10. *Pointed out in caption of Figure 15 that we only show the core.*

R1-8.2 In my opinion, several aspects of the notation are somewhat unfortunately chosen:

1. First, the symbol $M$ is strongly tied to "mass" for most readers, but the authors use it for mass divided by the density and unit footprint area, i.e., essentially for a length or depth. This is an unnecessary source of confusion for the readers.

2. Another issue, also arising in connection with $M$, is the use of a dot to denote a rate: It is well established tradition to use the dot to designate a time derivative. But the quantity $\dot{M}_{\Phi\to\Pi}$, e.g., is not the time derivative of some distinct dynamical quantity in the theory but the mass (or rather volume) flux across the boundary between the dense core and the suspension layer. Often used symbols for such fluxes are $Q$ (which might evoke heat, however) and $J$. This issue becomes particularly visible in Eq. (10), where $\dot{Q}_m$ and $q_{\Phi\to\Lambda}$ both designate heat fluxes.

3. The authors use $\Delta T$ and $\Delta D$ for the temperature and snow-depth lapse rates. The former has units °C/m, the latter is dimensionless. At least 99% of the readers will immediately interpret the expression $\Delta T$ as meaning the difference between two temperatures, again with units °C, rather than the intended lapse rate. Therefore, I suggest choosing a different symbol, say $a$, with subscripts $T$ or $d$.

4. In Eqs. (5), (6), (8), (17), (18), and (20) and also in the in-line formulas in lines 200 and 258–261, there are superfluous brackets or parentheses around fractions. This is not strictly wrong, but it does not add clarity, just clutter. Moreover, the use of parentheses and brackets appears haphazard. Most authors use brackets only if there already are parentheses in the expression within the brackets.

5. The choice of the double strike letter D in Eq. (7) appears poorly motivated and looks strange, in my opinion. Either use a normal D or give a convincing explanation for your choice.

Thank you for providing this valuable perspective. We agree that the current choice of symbols is less than ideal. We will ensure to find a new solution that remains consistent with the notations used in the existing RAMMS literature. Additionally, we will invest more time in providing a clear explanation of the symbols, possibly through an overview table, and make adjustments to improve clarity where possible.

   ***Overall:*** *we added overall in the manuscript more detailed explanations and adjusted the symbols:*

1. *We show now all equations depth H depended*

2. *$\Delta D$ and $\Delta T$ are now $\nabla D$ and $\nabla T$*

3. *Removed the brackets*

4. *We use now the $\mathcal{D}(x, y, z)$*

> R1-8.3 Besides the major issue of reference selection, the reference list also needs careful improvement with respect to content and formatting: Some references are listed twice, in some cases the journal name or the page numbers are missing, or article titles are poorly formatted. Please add DOIs where available (this is the case for most articles except ISSW proceedings, where a URL can be supplied instead).

We will clean this up.

> R1-8.4 Unless this is taken care of by Copernicus during copy editing and typesetting, the authors need to carefully review the manuscript, distinguishing between hyphen '-', en-dash '—', em-dash '—' and minus sign '−'. Physical units should be written with upright letters, and consistent and correct spacing between the number and units should be applied.

We will review the guidelines again and make the necessary adjustments.

**Language**

We will implement all proposed changes to language, grammar, punctuation and almost all proposed details to be added. For brevity, we only list and comment the few other recommendations herein:

- Adjustment of the naming of locations and avalanche paths according to Swisstopo names

- Add a table characterising the observed avalanches with maximum powder height etc

**Language**

**References**

Davidson, P.: Turbulence: An Introduction for Scientists and Engineers, Oxford University Press, Oxford, UK, 2nd edn., doi: 10.1093/acprof:oso/9780198722588.001.0001, 2015.

Issler, D., Jenkins, J., and Mcelwaine, J.: Comments on avalanche flow models based on the concept of random kinetic energy, Journal of Glaciology, 64, doi: 10.1017/jog.2017.62, 2017.

Issler, D., Gauer, P., Tregaskis, C., and Vicari, H.: Structure of equations for gravity mass flows with entrainment, Nature Communications, 15, doi: 10.1038/s41467-024-48605-6, 2024.

Zhuang, Y., Piazza, N., Xing, A., Christen, M., Bebi, P., Bottero, A., and Bartelt, P.: Tree blow-down by snow avalanche air-blasts: dynamic magnification effects and turbulence, Geophysical Research Letters, 50, e2023GL105 334, doi: 10.1029/2023GL105334, 2023.

---

## Referee Report (RR1)

All Line numbers given by me refer to the track changes document!

Given that the first answers are on the style of "we we can do this or that", I expected much more consise answers of what you acually did. But, most of the answers are still the same vague replies making finding the changes very difficult and cumbersum. I was not able to find all the answers to my first questions. Please carefully check my first review, and discuss with you co-authors if they are properly captured.

Most of my comments to improve discussion of subsection 4.3 and 4.4 are not answered or done. Please discuss them properly.

I find the story of the manuscript still rather difficult to follow. And the order of informations is not the best. In general, I have the feeling, that the experienced co-authors were not much active during the reviewing of the manuscript and too much effort is therefore handed over to the external reviewers.

The publication has definitively improved and has potential, but still needs some more effort. Please help the reviewers by more clearly state what you changed and where.

R2-2: OK.

R2-3: The suggested discussion of the simplification that alpha is temperature independent is missing. The authors only state it's constant value for cold and warm avalanche without clearifying what cold and warm means.

R2-4: I can not find the updated part in the "first paragraph on avalanche model" regarding the VdlS calibration. I see one sentence clarifying the calibration process at the very end of the introduction, however, to the reader not familier with the VdlS topography, the sentence is rather confusing. When summarizing the current state of the model from literature (e.g. R2-2), I would like to see also a paragraph about the calibration.

R2-5: Thank you for providing the observed avalanches of the last 60 years. Obviously, not most avalanches reach the road, but many (if not most) avalanches reach the valley flow, e.g. the flat part. Now I can better understand why you choose the idealize plane topography.

I am still missing the distances along the idealize plane topography (you give elevation and horizontal distance), but your verification works on the distances.
Naming: idealize plane (e.g. Fig 4) or inclined plane (e.g. Fig 8). Stick to one.

Regarding your definition of avalanche length or runout distance using the longest distance between two points in the affected area: this is a rather unusual definition, but it seem to work on most avalanches as they are much longer than wide. However, please add 1 or 2 sentences explaining this (and the limitation).

R2-7: I can not find the mentioned train of thoughts in the publication. I think, that discussion not using the VdlS data is important.

(BTW: VdlS has the most complete dataset in the world, your argument with SNOWPACK is of secondary importance and should not be mentioned in the discussion.)

Other comments:

- Line 90: Add reference to your supplementary material

- Line 98: The name of you institute is not correct, but is also not important here. Please remove.

- Line 148 and Line 138: Duplication of sentences. Please reread and revise the text between them!

- Fig 8: remove the points for warm temperatures. You can not show data you are not convinced of, and tell the people to take caution. Just dont show wrong / untested results.

- Line 515: How does the newly added subsection 4.2 on release length / angle and this sentence work together? From 4.2, I understand that only for very steep release areas the powder cloud goes further than the dense part (which is rather counter-intuitive, please discuss in more detail, e.g. what is terrain friction?)

- Line 522: The comparison between estimated front or approach velocity and maximum peak field velocities does not work! Please compare the velocities of Fig2 to equivalent values (extract the approach distances from the model run at the corresponding timesteps). See below to R2-32.

Fig14: Color scale of pressure has only one value and the wrong unit

- Line 523ff: I find the comparison with volumes from the drone data rather difficult to follow. Please rephase and explain to the reader what is important. (E.g. what is the difference you want to explain in Line 531?). Fig12 indicates core volume up to 120.000m^3 and erosion volumes up to 80.000m^3, but in Line 529 is only 30.000m^3 deposit. On the other hand, the drone data looks rather like no deposit at all (inside the outline to left and right of the outline has a fairly similar coloring and pattern). Can you explain this more in depth?

Fig 3: One can see that there has been significant erosion in the main part of the track (blue colors are basically no/only little snow left), however, in the deposition area most deposition is at 1m which equates to the normal snow cover prior to the event. Where do you expect is all the mass from the track? Please explain or mention this discrepancy.

- Line 538ff: And what does this mean for the simulation? Is the simulation now bad?

R2-16: I doubt there is a commonly accepted way how to estimate the snow temperature for avalanche dynamics purposes. It would be helpful for the reader to include one sentence to mention the discrepancy between the temperature in the release zone and the nearby flat field station measurements.

R2-29: I dont see the mark at 0.3C/100m. And line 407 shows you changed from 0.3 to 0.5C/100m. Please revise and be consistent.

R2-30: The sentences still says that a deep snowcover is need for the longevity of PSAs, and your answers suggests that you drive the erodible layer with the new snow. So, please rephrase the sentence and clearify. Also to R2-31: You should present the reader also with the limitations, and that the approximation fails for extreme avalanches that basically erode everything.

Fig10: Unfortunately, the modification of the figure removed the initial snow temperature of the mountain. Can this broad back?
Line 450: "increase by 3C from -8C to -2C" does not work...

R2-32: Well, ususally the questions from the reviewer are pointing to some missing information which could be of interest to the general reader and not only to the reviewer alone.
So, you are able to extract some velocities of the avalanche in Fig. 2, now here you show the frontal evolution over time. You can estimate a velocity from this plot inbetween the 5s steps, and you have another measurement to compare the simulation against, e.g. in
line 520ff.

R2-33: Adding half a sentence is not a discussion of other literatur. Please extend the discussion the literature mentioned in the first review round!
Similar holds true for R2-35.

R2-37: OK, you estimate some velocities from the video, but you dont use this information. As mentioned earlier, please add the front velocity to Fig.14 or to a similar figure.

R2-40: I can not find the part in the manuscript where you comment / discuss on the issue. Please refine and clearly point to the part in the manuscript

R2-40: You refuse to either redo the simulations, or to properly discuss the given question. I feel like this is one of the main points of the paper, but not at all discussed.

Fig15 (comment F16): you did not annotate the arm.

Line 533: Well, the answer you gave in the review manuscript should be given in the paper manuscript. You have to tell the reader what you want to say. Simply say we exclude this and that opens more questions that is helpful in any way.

Line 535: Still unclear.

Line 410: This sentence is still unexpected and does not make much sense without additional information. Please write with a couple of sentences this information and what implication this wind scour involves.

Line 580ff: The last paragraph in the Conclusion should go to Outlook (If one needs an outlook explicitly, better move those thoughts to the discussion).

Acknowlegments need update to state review by Dieter Issler and anonymous.

---

## Author Response (AR2)

**Response to Referees**
**Simulation of cold powder avalanches considering daily snowpack and weather situations to enhance road safety**

Julia Glaus, Katreen Wikstrom Jones, Perry Bartelt, Marc Christen, Lukas Stoffel, Johan Gaume, and Yves Bühler

March 5, 2025

Dear Editor,

We greatly appreciate the opportunity to proceed to the next round of the review process. We are thankful for the detailed and insightful feedback provided by the reviewers, which has been invaluable in improving the quality of our manuscript. In response to the feedback, we have copied the reviewers' comments into the blue boxes and gave our responses under each comment.

Sincerely,

Julia Glaus & co-authors

**Response to Reviewer R1**

**0.1 General assessment of the revised version**

The authors have clearly made a substantial effort in revising their manuscript, taking the criticism and suggestions of both reviewers seriously. The text is much better balanced and easier to read than before (a few exceptions will be mentioned below). Some figures without particular value for the paper were removed, others that have a specific purpose were added, and many others were improved to make them more informative or easier to understand.

**0.2 Remaining issues**

> The description of RAMMS::EXTENDED has become more self-contained and easier to understand. However, the more detailed description of the closure assumptions has also brought additional problems to light that must be dealt with. The main problem areas are the following:

> C1: In the balance of internal energy (directly linked to the temperature of the avalanche core), the source term $\dot{Q}_m$ describes the heat loss due to melting of snow. Equation (11) specifies the form of $\dot{Q}_m$ in terms of an integral over an infinitesimal time interval, which gives an infinitesimally small result if $\dot{Q}_m$ is finite. Moreover, the integration interval $dt$ does not appear on the right-hand side of the equation. This expression is presumed proportional to the temperature difference between the avalanche core and the melting point of ice, $T_\Phi - T_m$. Now, the temperature of a dense or fluidized flow is lower or at most equal to $T_m$, hence there is no heat flux available for melting. (In a very dilute flow, i.e., in the powder cloud, $T_\Phi > T_m$ is possible in principle.) As far as I can see, the problem arises because only the macroscopic quantity $T_\Phi$ is available, while the melting is driven by local heat production, i.e., at the mesoscopic or even microscopic level. Therefore, some assumptions on the link between the macroscopic variables and the mesoscopic ones are needed, but the scheme employed by the authors is not suitable.
>
> In my opinion, the simplest and best solution to this conundrum is to remove the description of melting from the paper because this specific term was not used in this work focusing on cold powder snow avalanches. The problem raises, however, some questions concerning the correctness of papers on wet-snow avalanches published earlier.

We agree to the review comment on the inconsistency of the description of the melting process. As this publication focuses on cold avalanches, we removed this part at the very end of Chapter 3.1 and added a short statement, that more information on the melting process can be found in (Zhuang et al., 2023).

C2: The authors provide an argument why the gravitational term in the momentum balance for the powder cloud is not important most of the time: The momentum injected into it by the mass ejected from the dense core is said to be much larger than the gravitational pull,

$$\dot{H}\,\Phi \rightarrow \Pi\,u_\Phi \gg \left(1 - \frac{\rho_\Lambda}{\rho_\Pi}\right) G_\Pi\,.$$

However, this equation is dimensionally inconsistent; the right-hand side should be multiplied by $H_\Pi$. This can be a large number, hence the gravitational term does become important as soon as the powder cloud has developed sufficiently.

We have corrected the misspelling and added $H_\Pi$ to the right-hand side of the equation. We agree that for larger, fully developed powder clouds, this assumption may become critical. Consequently, we have included a discussion in line 311 of the track-changes file to highlight the limitations of these assumptions.

C3: The explanation of the entrainment model remains difficult to understand, possibly due to misunder-standings of the mechanisms at work in erosion and entrainment. Another problem is that Eq. (21) is not positive definite, contrary to the authors' statement. It is unclear whether the code enforces positivity of the erosion coefficient $\kappa$ or not. At any rate, if $\kappa$ becomes 0 below a limiting slope angle, erosion during run-out in level terrain or even in counter-slopes cannot be described, even though it has been observed many times.

In the code of RAMMS::EXTENDED, the positivity of the $\kappa$ coefficient is enforced. We added a comment from line 346 on in the track-changes file on the limitation of the system and mentioned also the case of VdlS 1999 where exactly the case of erosion in the flat terrain / counter slope was observed.

While the authors have added some suggested references to work by other researchers, still well over half of the references are to work carried out at SLF. Most of these references are not out of context, but not all of them are clearly needed. Such a high degree of self-citation in an international research field like snow avalanches is highly unusual and leaves readers with a poor impression. I suggest the authors prune their self-citations by about half.

We agree that this is an unfortunate situation. As the first half of the publication summarizes the individual components of the simulation tool RAMMS::extended, a large number of in-house citations naturally arise, as this tool was developed over many years at SLF. We tried to add more citations.

**0.3 Minor Remarks**

Please see the annotated manuscript for numerous small corrections and suggestions concerning the language and presentation. In particular, the reference list needs some further corrections and additional bibliographic information.

We worked through all numerous small corrections and adjust the text accordingly to improve the language and presentation.

**0.4 Recommendation to the editor**

In view of the major improvements of the manuscript and the nature of the remaining issues, I recommend to the editor to accept the paper for publication in NHESS on the condition that the points listed above are satisfactorily resolved in a minor revision.

**Response to Reviewer R2**

**General Comments**

Given that the first answers are on the style of "we we can do this or that", I expected much more consise answers of what you acually did. But, most of the answers are still the same vague replies making finding the changes very difficult and cumbersum. I was not able to find all the answers to my first questions. Please

carefully check my first review, and discuss with you co-authors if they are properly captured. Most of my comments to improve discussion of subsection 4.3 and 4.4 are not answered or done. Please discuss them properly. I find the story of the manuscript still rather difficult to follow. And the order of informations is not the best. In general, I have the feeling, that the experienced co-authors were not much active during the reviewing of the manuscript and too much effort is therefore handed over to the external reviewers. The publication has definitively improved and has potential, but still needs some more effort. Please help the reviewers by more clearly state what you changed and where.

C4: R2-3: The suggested discussion of the simplification that alpha is temperature independent is missing. The authors only state it's constant value for cold and warm avalanche without clearifying what cold and warm means.

We added the definition from (Köhler et al., 2018) in line 235, in the track-changes file.

C5: R2-4: I can not find the updated part in the "first paragraph on avalanche model" regarding the VdlS calibration. I see one sentence clarifying the calibration process at the very end of the introduction, however, to the reader not familier with the VdlS topography, the sentence is rather confusing. When summarizing the current state of the model from literature (e.g. R2-2), I would like to see also a paragraph about the calibration.

According to personal communication with Perry Bartelt, the simulation tool was originally calibrated based on the friction parameters of RAMMS::Avalanche. During the development of RAMMS::EXTENDED, it underwent continuous calibration by incorporating new features and simulating numerous avalanches. In the track-changes file we added more details in line 66, which now reads as follows: "The model was calibrated using avalanches observed at Vallée de la Sionne (VdlS) (Ammann, 1999), considering only those that did not reach the counter slope according to (Bartelt, 2025). Additionally, avalanches from winter 1999 in Switzerland were used which are presented in Vallet et al. (2001). "

C:6 R2-5: I am still missing the distances along the idealize plane topography (you give elevation and horizontal distance), but your verification works on the distances. Naming: idealize plane (e.g. Fig 4) or inclined plane (e.g. Fig 8). Stick to one. Regarding your definition of avalanche length or runout distance using the longest distance between two points in the affected area: this is a rather unusual definition, but it seem to work on most avalanches as they are much longer than wide. However, please add 1 or 2 sentences explaining this (and the limitation).

We added the runout distance scale in the graphic as you described. Overall, we adjusted the naming to "idealized plane." In the last round of discussion we added the limitations of our methods in line 125 of the track-changes file: "While this method works well for simple avalanches, it requires careful consideration in cases where avalanches exhibit finger formation or the avalanche strongly deviates in the lateral direction. Additionally, the code accounts for the case of perfectly symmetric avalanches (as observed on idealized slopes) by measuring the distance between the two outermost points in the flow direction. "

C7: R2-7: I can not find the mentioned train of thoughts in the publication. I think, that discussion not using the VdlS data is important. (BTW: VdlS has the most complete dataset in the world, your argument with SNOWPACK is of secondary importance and should not be mentioned in the discussion.)

We decided not to include it, as reusing avalanches to analyze a simulation tool that was already calibrated with them did not seem appropriate. Additionally, in the Brämabühl area, the avalanches have a long runout, making it a suitable setup to investigate how sensitive the avalanche runout distance is to the input data. In contrast, the VdlS avalanches run into the counter slope. At this point, the RAMMS model reaches its limitations, as described in more detail in the paper (see the erosion model chapter line 346 of the track-changes file). However, we included one VdlS avalanche that was not part of the calibration process to further validate our approach, as mentioned in line 114 of the track-changes file.

**Other Comments**

C8:

1. Line 90: Add reference to your supplementary material

2. Line 98: The name of you institute is not correct, but is also not important here. Please remove

3. Line 148 and Line 138: Duplication of sentences. Please reread and revise the text between them

4. Fig14: Color scale of pressure has only one value and the wrong unit

1. References will be added as soon as the supplementary material is published.

2. We removed the institute name.

3. We removed the sentence and rewrote the section according to the review inputs in the additional document from Dieter Issler.

4. We adjusted the color scale and value and added a discussion on the powder cloud values.

C9: Fig 8: remove the points for warm temperatures. You can not show data you are not convinced of, and tell the people to take caution. Just dont show wrong / untested results.

In the text we mention the limitations of the model for warm avalanches. We think it is important to present the full range of parameter variations, to illustrate the numerical behavior of the simulation, even though it is not fully tested.

C10: Line 515: How does the newly added subsection 4.2 on release length / angle and this sentence work together? From 4.2, I understand that only for very steep release areas the powder cloud goes further than the dense part (which is rather counter-intuitive, please discuss in more detail, e.g. what is terrain friction?)

We agree that the term 'friction' is misleading in this context. We have adjusted it to 'terrain roughness,' referring to features such as rocks, vegetation, and old snow structures. Accordingly, we have rephrased the sentence in Line 452 as follows: "On a real terrain, the terrain roughness (cased e.g. by vegetation and rocks) would strongly decelerate the avalanche core and reduce its runout distance."

C11: Line 522: The comparison between estimated front or approach velocity and maximum peak field velocities does not work! Please compare the velocities of Fig2 to equivalent values (extract the approach distances from the model run at the corresponding timesteps). See below to R2-32.

We have corrected the evaluation and present the simulated and measured values of the powder cloud front in the newly added Table 3. As stated in the caption, we only assess whether the front velocity falls within the correct order of magnitude. The purpose of this comparison is to verify that the results lie within a reasonable range rather than to imply that the simulation accurately captures velocity at a highly detailed level.

C12: Line 523ff: I find the comparison with volumes from the drone data rather difficult to follow. Please rephase and explain to the reader what is important. (E.g. what is the difference you want to explain in Line 531?). Fig12 indicates core volume up to 120.000 $m^3$ and erosion volumes up to 80.000 $m^3$, but in Line 529 is only 30.000 $m^3$ deposit. On the other hand, the drone data looks rather like no deposit at all (inside the outline to left and right of the outline has a fairly similar coloring and pattern). Can you explain this more in depth?

We have rephrased the paragraph to provide a more detailed explanation of the calculation for the comparison between the simulated and measured deposition. It is not possible to directly compare the values in Figure 12 with the deposition data calculation, as Figure 12 considers the mass of the entire avalanche, whereas this paragraph focuses solely on the deposition in the runout zone due to gaps in data in the forested area. We have included this clarification in the text to avoid any misunderstanding. Additionally, for the drone data, we have specified that the calculation takes into account the compression of the old snow by the heavy avalanche deposit, proportional to the density increase up to $450\,kg/m^3$.

C13: Fig 3: One can see that there has been significant erosion in the main part of the track (blue colors are basically no/only little snow left), however, in the deposition area most deposition is at 1m which equates to the normal snow cover prior to the event. Where do you expect is all the mass from the track? Please explain or mention this discrepancy.

As answered in comment C12, the avalanche compressed the old snow cover. Additionally, the color scale shows all snow height values above 2m in red, and not a fixed limit at 2m.

C14: Line 538ff: And what does this mean for the simulation? Is the simulation now bad?

This statement in the manuscript concerns more the approach of averaging local station measurements and deriving conclusions about the conditions in a single avalanche track. With this observation, we show that even though the three avalanche tracks are very close to each other, they were still exposed to different wind influences, and consequently, to varying amounts of wind-drifted snow. Further research on this topic is currently being conducted in the study by (Ruttner-Jansen et al., 2023), where two measurement stations were placed directly in the release areas at Brämabühl to evaluate this approach. We hope to publish a more detailed analysis of this topic next season.

C15: R2-16: I doubt there is a commonly accepted way how to estimate the snow temperature for avalanche dynamics purposes. It would be helpful for the reader to include one sentence to mention the discrepancy between the temperature in the release zone and the nearby flat field station measurements.

We answered this in our statement in comment C14.

C16: R2-29: I dont see the mark at 0.3C/100m. And line 407 shows you changed from 0.3 to 0.5C/100m. Please revise and be consistent.

We adjusted the parameters. As the graphic is already quite overloaded, we decided not to add more information.

C17: R2-30: The sentences still says that a deep snowcover is need for the longevity of PSAs, and your answers suggests that you drive the erodible layer with the new snow. So, please rephrase the sentence and clearify. Also to R2-31: You should present the reader also with the limitations, and that the approximation fails for extreme avalanches that basically erode everything.

- The approach of just taking the new snow layer into account is part of the traditional hazard mapping scheme. Further research is done in the ISSW publication (Glaus et al. 2024) where additional weaklayers are taken into account as described also in the outlook section.

- Limitation of the erosion model: We fixed this discussion according to Dieter Isslers review comments (see C3).

C18: Fig10: Unfortunately, the modification of the figure removed the initial snow temperature of the mountain. Can this broad back? Line 450: "increase by 3C from -8C to -2C" does not work...

The initialization is already presented in Figure 5, and we have removed it here to avoid redundancy but added a comment in the caption. Additionally we fixed the typo in the temperatures.

C19: R2-32: Well, ususaly the questions from the reviewer are pointing to some missing information which could be of interest to the general reader and not only to the reviewer alone. So, you are able to extract some velocities of the avalanche in Fig. 2, now here you show the frontal evolution over time. You can estimate a velocity from this plot inbetween the 5s steps, and you have another measurement to compare the simulation against, e.g. in line 520ff.

As answered in C11 we added the comparison in Table 3 and discussed it in the text. We did the comparison to the simulation with dump steps of 1 s to evaluate the velocity inbetween the pictures.

C20: R2-33: Adding half a sentence is not a discussion of other literatur. Please extend the discussion the literature mentioned in the first review round! Similar holds true for R2-35.

As we do not discuss the relation between erosion and frontal velocity in this publication, we also do not include an extended literature review about this topic.

C21: R2-37: OK, you estimate some velocities from the video, but you dont use this information. As mentioned earlier, please add the front velocity to Fig.14 or to a similar figure.

As answered above in C11, we added the comparison in Table 3.

C22: R2-40: I can not find the part in the manuscript where you comment / discuss on the issue. Please refine and clearly point to the part in the manuscript. R2-40: You refuse to either redo the simulations, or to properly discuss the given question. I feel like this is one of the main points of the paper, but not at all discussed.

We agree that it is difficult to explain exactly why the longest avalanches are simulated for a certain release temperature. However, in this publication, we use RAMMS::EXTENDED as calibrated by the developers of RAMMS and evaluate how this calibration performs for the Brämabühl avalanches. The exact location of this peak is not influenced by the authors of this publication. We conducted the same simulations for the Salezer avalanche and one VdlS avalanche from 2016, and observed the same peak. Therefore, rerunning the simulations, which have already been performed multiple times over the last two years, will not alter the results.

C23: Fig15 (comment F16): you did not annotate the arm.

We have revised the paragraph, as mentioned in comment C12 above, regarding the deposition. We can only provide a qualitative description, noting that higher deposition occurs within roughly the same region. However, since there is no data available on the snow cover prior to the avalanche, it is possible that an old avalanche deposit was already present. Furthermore, we aim to avoid giving the impression that the simulation is precise enough to replicate the avalanche with such a high level of detail.

C24: Line 533: Well, the answer you gave in the review manuscript should be given in the paper manuscript. You have to tell the reader what you want to say. Simply say we exclude this and that opens more questions that is helpful in any way.

We added those information as described in comment C12.

C25: Line 535: Still unclear.

We rephrased the paragraph as described in comment C12.

C26: Line 410: This sentence is still unexpected and does not make much sense without additional information. Please write with a couple of sentences this information and what implication this wind scour involves.

We are not sure which passage you refer to as line 410 is about the definition of the random kinetic energy.

C27: Line 580ff: The last paragraph in the Conclusion should go to Outlook (If one needs an outlook explicitly, better move those thoughts to the discussion).

We adjusted the location of the paragraph as suggested.

**References**

Ammann, W. J.: A new Swiss test-site for avalanche experiments in the Vallée de la Sionne/Valais, Cold Regions Science and Technology, 30, 3–11, doi: 10.1016/S0165-232X(99)00010-5, 1999.

Bartelt, P.: Personal communication, 2025.

Köhler, A., Fischer, J. T., Scandroglio, R., Bavay, M., McElwaine, J., and Sovilla, B.: Cold-to-warm flow regime transition in snow avalanches, The Cryosphere, 12, 3759–3774, doi: 10.5194/tc-12-3759-2018, 2018.

Ruttner-Jansen, P., Glaus, J., Wieser, A., and Buehler, Y.: A Measurement System for Mapping Snow Distribution Changes in an Avalanche Release Zone, URL `https://arc.lib.montana.edu/snow-science/objects/ISSW2023_O10.05.pdf`, 2023.

Vallet, J., Gruber, U., and Dufour, F.: Photogrammetric avalanche volume measurements at Vallée de la Sionne, Switzerland, Annals of Glaciology, 32, 141–146, doi: 10.3189/172756401781819689, 2001.

Zhuang, Y., Piazza, N., Xing, A., Christen, M., Bebi, P., Bottero, A., and Bartelt, P.: Tree blow-down by snow avalanche air-blasts: dynamic magnification effects and turbulence, Geophysical Research Letters, 50, e2023GL105 334, doi: 10.1029/2023GL105334, 2023.